# Key phosphorylation sites for robust β-arrestin2 binding at the MOR revisited
Owen Underwood[1,2], Sebastian Fritzwanker[3], Jaqueline Glenn[1,2], Nina Kathleen Blum[3], Arisbel Batista-Gondin[4], Julia Drube [5], Carsten Hoffmann[5], Stephen J. Briddon[1,2], Stefan Schulz [3,6] & Meritxell Canals [1,2] ✉

Desensitisation of the mu-opioid receptor (MOR) is proposed to underlie the initiation of opioid analgesic tolerance and previous work has shown that agonist-induced phosphorylation of the MOR C-tail contributes to this desensitisation. Moreover, phosphorylation is important for β-arrestin recruitment to the receptor, and ligands of different efficacies induce distinct phosphorylation barcodes. The C-tail [370]TREHPSTANT[379] motif harbours Ser/Thr residues important for these regulatory functions. [375]Ser is the primary phosphorylation site of a ligand-dependent, hierarchical, and sequential process, whereby flanking [370]Thr, [376]Thr and [379]Thr get subsequently and rapidly phosphorylated. Here we used GRK KO cells, phosphosite specific antibodies and site-directed mutagenesis to evaluate the contribution of the different GRK subfamilies to ligand-induced phosphorylation barcodes and β-arrestin2 recruitment. We show that both GRK2/3 and GRK5/6 subfamilies promote phosphorylation of [370]Thr and [375]Ser. Importantly, only GRK2/3 induce phosphorylation of [376]Thr and [379]Thr, and we identify these residues as key sites to promote robust β-arrestin recruitment to the MOR. These data provide insight into the mechanisms of MOR regulation and suggest that the cellular complement of GRK subfamilies plays an important role in determining the tissue responses of opioid agonists.

Mu-opioid receptor (MOR) agonists, such as morphine, remain among the most effective drugs for the relief of acute severe pain. Unfortunately, prolonged use of opioid agonists results in the development of analgesic tolerance, which requires dose escalations to maintain an analgesic effect. Despite its clinical relevance, the molecular processes resulting in the development of analgesic tolerance upon chronic exposure to opioids are not fully understood, although it is clear that it is a complex process that entails multiple adaptations both at the cellular level and at the level of neuronal circuitry.

At a cellular level, MOR desensitisation is a recurrent mechanism proposed to underlie the initiation of tolerance. Such desensitisation is driven by phosphorylation of Ser and Thr residues in the receptor carboxyl-terminal[1–3], which drives subsequent recruitment of β-arrestins to the receptor[4]. Indeed, mutation of all potential phosphorylation sites within the MOR C-tail results in a dramatic reduction of MOR acute desensitisation in model cell systems[1,3] and brain slices[5] as well as enhanced analgesia and loss of analgesic tolerance in vivo[5].

The MOR C-tail contains 11 Ser/Thr residues. Phosphosite-specific antibodies and quantitative mass spectrometry (MS) have identified two specific motifs, [354]TSST[357] and [370]TREHPSTANT[379], that undergo agonist-induced, sequential, and hierarchical phosphorylation[6–10]. Within the [370]TREHPSTANT[379] motif, the middle residue, Ser[375], is proposed to be the primary phosphorylation site, which is phosphorylated upon activation of MOR with all opioid agonists[9]. Low efficacy agonists such as morphine or oliceridine induce selective phosphorylation of Ser[375] without further phosphorylation of other residues within the [354]TSST[357] or [370]TREHPSTANT[379] cassettes[9,10]. Such limited phosphorylation does not promote robust β-arrestin recruitment in HEK293 cells[4,9], nor does it facilitate receptor internalisation either in HEK293 cells or rat cortical neurons[4,9,11]. In contrast, high efficacy agonists, including DAMGO and

[1]Division of Physiology, Pharmacology and Neuroscience, School of Life Sciences, Queen's Medical Centre, University of Nottingham, Nottingham, UK. [2]Centre of Membrane Proteins and Receptors (COMPARE), Universities of Nottingham and Birmingham, Birmingham, Midlands, UK. [3]Institut für Pharmakologie und Toxikologie, Universitätsklinikum Jena, Friedrich-Schiller-Universität Jena, Jena, Germany. [4]Drug Discovery Biology Theme, Monash Institute of Pharmaceutical Sciences, Monash University, Victoria, Australia. [5]Institut fur Molekulare Zellbiologie, CMB – Center for Molecular Biomedicine, Universitätsklinikum Jena, Friedrich-Schiller-Universität Jena, Jena, Germany. [6]7TM Antibodies GmbH, Hans-Knöll-Straße 6, D-07745 Jena, Germany. ✉e-mail: m.canals@nottingham.ac.uk

fentanyl, drive higher order phosphorylation, initiated by modification of Ser[375] and propagated to the flanking residues Thr[370], Thr[376] and Thr[379] in a hierarchical phosphorylation cascade that in turn promotes both β-arrestin recruitment and robust receptor internalization in HEK293 cells[4,6,9,10]. However, while the relevance of the [370]TREHPSTANT[379] cassette is clear, the relative contribution of the individual Ser/Thr residues, beyond Ser[375], remains to be determined. Finally, while Ser[356] and Thr[357] undergo agonist-induced phosphorylation, mutation of the complete [354]TSST[357] motif does not affect desensitisation or internalisation of the MOR in model cell systems[3,4], thus, the relevance of this motif for MOR regulation remains unclear.

G protein receptor kinases (GRKs) are the main mediators of GPCR phosphorylation. In humans, the GRK family consists of seven subtypes, GRK1-7. Of these, GRK1 and GRK7 are specifically expressed in the retina, and GRK4 is predominantly expressed in specific tissues such as the testis or the heart. The remaining four subtypes (GRK2, GRK3, GRK5 and GRK6) are widely expressed in the body[12] and are classified into the GRK2/3 and GRK5/6 subfamilies[13]. These two subfamilies have different subcellular localisation and activation mechanisms. GRK2/3 reside in the cytosol and translocate to the plasma membrane upon binding of Gβγ subunits, released upon GPCR activation and heterotrimeric G protein dissociation. In contrast, GRK5/6 are membrane tethered (via a PIP$_2$ binding domain or palmitoylation, respectively) and do not bind Gβγ subunits.

The contribution of specific GRKs and other kinases (e.g., PKC) to the phosphorylation of MOR has been extensively studied[14–19]. Experiments using pharmacological inhibitors (e.g. Compound 101, selective for GRK2/3) or si/shRNAs for specific isoforms, have suggested that the action of GRK2/3 underlies the higher-order phosphorylation induced by high efficacy agonists, and that GRK5/6 mediate phosphorylation of Ser[375] induced by morphine[9,11]. Multiple studies have also shown that overexpression of GRK2 overcomes the partial phosphorylation induced by morphine, and that overexpressed GRK2 facilitates multi-site phosphorylation, coupled with measurable β-arrestin recruitment and receptor internalisation[4,19,20]. However, despite all the accumulated evidence, direct assessment of the contribution of specific kinase isoforms to the ligand-induced MOR phosphorylation barcode is missing. This is partly due to the lack of tools with which to directly interrogate the role of each GRK family.

CRISPR/Cas9 technologies have been instrumental to address key mechanistic questions of GPCR signalling. HEK293 cells engineered to lack selected G proteins or β-arrestins have helped elucidate the contribution (or lack of thereof) of specific isoforms of these proteins to receptor signalling and regulation[21–24]. Along these lines, the recent generation of HEK293 cells lacking specific GRK isoforms[25–28] provide an invaluable means to assess the impact of isoform-specific phosphorylation barcodes to subsequent events of β-arrestin recruitment and receptor internalisation or further downstream signalling. A combination of G protein, β-arrestin and GRK knockout (KO) cells has recently been used to show that Gq activity acts as a molecular switch in GRK-subtype selectivity for the AT1R whereby the Gq heterotrimer acts as a negative modulator of β-arrestin by suppressing the function of GRK5/6, in turn enhancing the access of GRK2/3 to AT1R, which collectively affects the phosphorylation barcode of the receptor and determines β-arrestin function[26]. A combinatorial panel of GRK-KO cells also enabled the differentiation of GPCRs depending on their GRK-subtype dependence and demonstrated how selective engagement of specific kinases mediated the formation of distinct receptor-β-arrestin complex configurations[25]. In the context of the MOR, receptor internalisation and β-arrestin recruitment have been measured in HEK293 cells lacking GRK2 and/or GRK3 and confirmed the role of these isoforms in such processes[25,27]. However, the impact of these knockouts on receptor phosphorylation profiles as well as the role of GRK5 and GRK6 in agonist-driven regulation have not yet been directly investigated.

Here we used HEK293 GRK KO cells in combination with phosphosite specific antibodies and site-directed mutagenesis of the MOR to evaluate the contribution of the different GRK subfamilies to ligand-induced phosphorylation barcodes and β-arrestin2 recruitment. We show that while both

GRK subfamilies (GRK2/3 and GRK5/6) promote phosphorylation of Thr[370] and Ser[375], only GRK2/3 induce phosphorylation of Thr[376] and Thr[379], and we identify these residues as key sites to promote robust β-arrestin recruitment to the receptor. As MOR phosphorylation is a key determinant for receptor desensitisation, β-arrestin recruitment, and internalisation, the molecular determinants identified here provide insight into the mechanisms of MOR regulation and suggest that the cellular complement of the different GRK subfamilies plays an important role in determining the tissue responses of distinct opioid agonists.

## Results

### Agonist-induced MOR phosphorylation by distinct GRK families

Within the C-terminus of the MOR, the sequence [370]TREHPSTANT[379] comprises four phosphorylation sites (Thr[370], Ser[375], Thr[376] and Thr[379]) that collectively have been shown to be required for ligand-induced β-arrestin recruitment, as well as desensitisation and internalisation of the receptor[1,3,4]. Accumulated evidence also shows that high efficacy opioid agonists, such as DAMGO and fentanyl, induce robust phosphorylation of all four residues within this sequence (Fig. 1)[4,9,10]. In contrast, MOR agonists with lower efficacy, such as morphine, induce a more partial or incomplete phosphorylation pattern, with phosphorylation of only Ser[375] and Thr[376] and minimal to no-phosphorylation detected at Thr[370] and Thr[379] (Fig. 1)[4,9]. To investigate the contribution of the different GRK families to such ligand-induced phosphorylation profiles, we determined the phosphorylation state of the MOR when expressed in CRISPR-Cas9 edited HEK293 cells lacking GRK2/3/5/6 expression (ΔQ-GRK) or deficient in one of the GRK isoform families (ΔGRK2/3 and ΔGRK5/6)[25]. As expected, knock-out of all GRKs expressed in HEK293 cells resulted in the absence of ligand-induced phosphorylation within the [370]TREHPSTANT[379] sequence (Fig. 1). When the MOR was expressed in cells lacking GRK2/3 (ΔGRK2/3), no phosphorylation of Thr[376] or Thr[379] could be detected for any ligand (Fig. 1). In contrast, differences between MOR agonists were apparent in cells lacking GRK5/6 (ΔGRK5/6) (Fig. 1). In these cells, no detectable phosphorylation in the [370]TREHPSTANT[379] sequence was observed upon stimulation with morphine, whereas stimulation with DAMGO and fentanyl induced phosphorylation of all four phospho-sites, albeit to a lesser extent than in control cells (Fig. 1).

Altogether, these results show that DAMGO- and fentanyl-induced phosphorylation of the two distal phospho-sites of the [370]TREHPSTANT[379] sequence (Thr[376] and Thr[379]) is mediated by GRK2/3 suggesting that the GRK5/6 subfamily is dispensable for the higher-order phosphorylation induced by these ligands.

### Contribution of GRK subfamilies to ligand-induced β-arrestin2 recruitment to the MOR

As C-tail phosphorylation is a key step for β-arrestin2 recruitment to the MOR, we investigated whether the distinct, ligand-dependent [370]TREHPSTANT[379] phosphorylation patterns elicited by individual GRK families impact the extent of β-arrestin2 recruitment. In control cells, DAMGO elicited the most robust recruitment of all ligands, while fentanyl induced 60% of the recruitment induced by DAMGO and morphine induced weak ( ~ 10%) β-arrestin2 recruitment to the MOR (Fig. 2A). The order of efficacies and potencies for the three agonists agrees with previous reports (Table 1)[20,29]. No significant β-arrestin2 recruitment was detected in ΔQ-GRK cells, in agreement with the lack of phosphorylation of the [370]TREHPSTANT[379] sequence and further supporting the key role of GRKs for β-arrestin2 interactions with the MOR (Fig. 2B). The recruitment induced by DAMGO, fentanyl and morphine in cells lacking GRK5/6 (ΔGRK5/6) was not different to that observed in control cells (Fig. 2D, Table 1). In contrast, in ΔGRK2/3 cells the extent of β-arrestin2 recruitment induced by DAMGO and fentanyl was significantly reduced, eliciting only 32% and 50% of the recruitment detected in control cells, respectively. Conversely, the recruitment induced by morphine in ΔGRK2/3 cells was 5-fold higher than that in control cells (Fig. 2C, Table 1). As expected, the ability of MOR to interact with G proteins, as measured by recruitment of

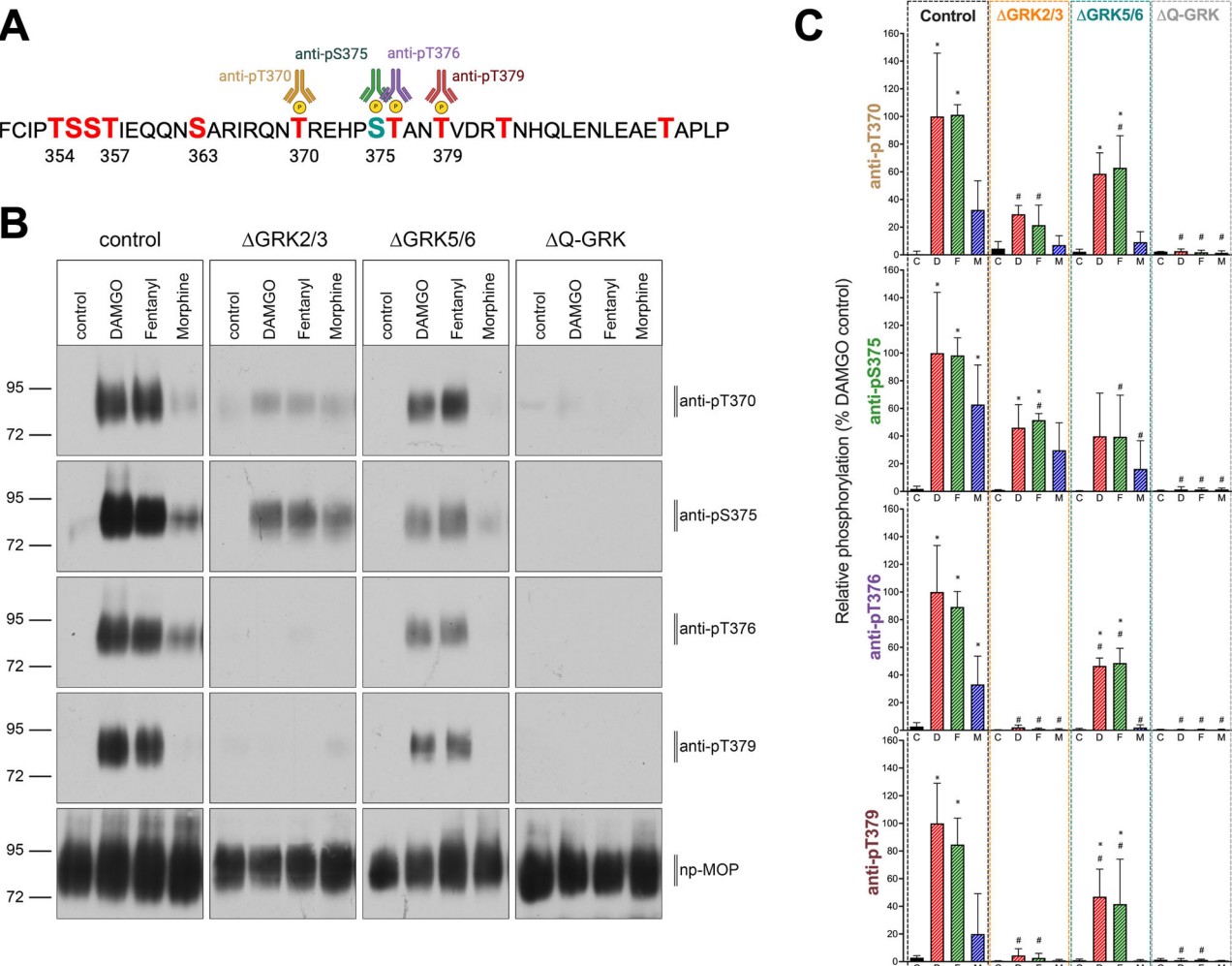

**Fig. 1 | MOR phosphorylation in control and GRK knockout cells. A** Sequence of the mouse MOR C-terminus. Ser/Thr highlighted as potential phosphorylation sites. Antibodies used in this study are shown. **B** Site specific phosphorylation of the MOR was detected in lysates of control, quadruple (ΔQ-GRK) or GRK2/3 (ΔGRK2/3) or GRK5/6 (ΔGRK5/6) subfamily knockout cells stably expressing HA-tagged MOR. Prior to immunoprecipitation with HA beads, cells were treated with DAMGO ($10^{-5}$ M), Fentanyl ($10^{-6}$ M) or Morphine ($10^{-5}$ M) for 30 min at 37 °C. Detection of site-specific phosphorylation was performed with phospho-site specific antibodies as described in Materials and Methods. Representative blot is shown (**C**) Quantification of Western Blots. Data show the mean ± SEM of 3 independent experiments. # Indicates $p < 0.05$ vs Control cell line, and * indicates $p < 0.05$ vs control (vehicle) treatment within cell line using one-way ANOVA (See Suppl Table 1 for detailed analysis).

mGsi, was not significantly affected in any of the knock-out cells (Supp Fig. 1, Supp Table 2). Moreover, the effects observed upon GRK knock-out did not correlate with differences in MOR expression in the different cell types (Supp Fig. 2A).

Together with the phosphorylation patterns described above, these data provide direct evidence supporting previous reports that suggested that GRK2 and GRK3 are the main GRK isoforms involved in the robust β-arrestin2 recruitment induced by DAMGO and fentanyl. Our data also reveal that the higher-order phosphorylation of Thr[376] and Thr[379] is required for such robust recruitment as partial modification of the [370]TREHPSTANT[379] sequence by phosphorylation of only Thr[370] and Ser[375] (as induced by morphine in control cells, or by all ligands in ΔGRK2/3 cells) results in a significantly reduced (but still concentration-dependent) β-arrestin2 recruitment.

### Rescued and enhanced β-arrestin2 recruitment upon over-expression of a single GRK isoform

GRK2 overexpression has often been used to enhance β-arrestin2 recruitment at the MOR and other GPCRs, facilitating the construction of concentration response curves for weak partial agonists. Indeed, we and others have previously shown that upon overexpression of GRK2 morphine induces higher-order phosphorylation of the MOR C-tail, concomitant with a robust increase in the ability of this ligand to induce β-arrestin2 recruitment[4,30] However, these previous observations were performed in cells with endogenous expression of GRKs, compounding the interpretation of the effect of overexpression of a single GRK isoform. To address this, we transfected ΔQ-GRK cells with individual GRK isoforms and measured agonist-induced β-arrestin2 recruitment to the MOR. Overexpression of GRK2, GRK3 and GRK5 in ΔQ-GRK cells not only rescued but enhanced β-arrestin2 recruitment by all agonists (Fig. 3, Table 2). The most robust effect was observed upon overexpression of GRK2 and GRK3, where all ligands induced maximal levels of recruitment with increases ranging from 4- to 10-fold the levels of recruitment observed in control cells. Interestingly, overexpression of GRK6 rescued the compromised signal produced by the absence of all GRKs, however, it did not lead to significant increases in β-arrestin2 recruitment above that observed in control cells (Fig. 3). The reduced recovery observed upon GRK6 overexpression was not due to an enhanced constitutive interaction between β-arrestin2 and MOR facilitated by this specific kinase, as the baseline upon GRK overexpression in ΔQ-GRK was similar in all conditions (Supp Fig. 3). Interestingly, the effect of

**Fig. 2 | β-arrestin2 recruitment to the MOR in control and GRK knockout cells.** β-arrestin2-Venus recruitment to the Flag-mMOR-Nluc upon stimulation with DAMGO, fentanyl or morphine for 10 min at 37 °C in (**A**) control cells, (**B**) quadruple (ΔQ-GRK) or subfamily (**C**) ΔGRK2/3 or (**D**) ΔGRK5/6 knockout cells. Data show the mean ± SEM baseline-corrected BRET ratio of 5 independent experiments performed in triplicate. The dotted lines in (**B**–**D**) represent the corresponding curves in control cells for comparison.

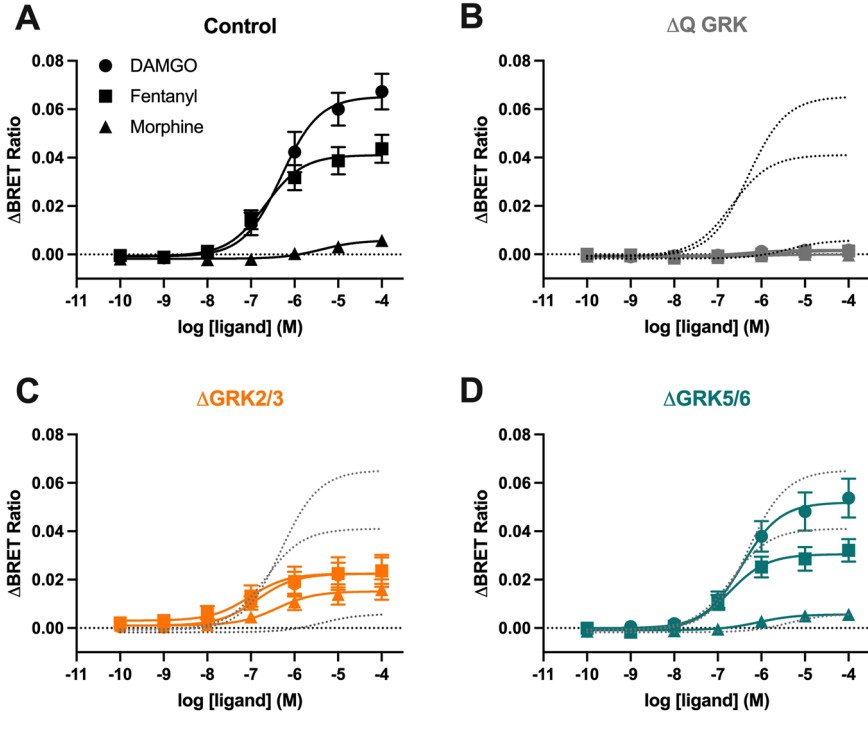

### Table 1 | Potency ($pEC_{50}$) and maximal effect ($E_{max}$) for DAMGO, morphine and fentanyl β-arrestin2 recruitment in control and GRK knock-out cells

| | | Control | ΔGRK2/3 | ΔGRK5/6 | QGRK |
|---|---|---|---|---|---|
| **DAMGO** | $pEC_{50}$ | 6.3 [6.7–5.9] | 6.9 [7.6–6.1] | 6.4 [6.9–6.0] | - |
| | $E_{max}$ (R, %C) | 0.07 (100) | 0.02 (32)* | 0.05 (79) | 0 (4)* |
| | $E_{max}$ (%) | 100 | 100 | 100 | - |
| **Morphine** | $pEC_{50}$ | 5.4 [6.3–4.5] | 6.4 [7.3–5.6]* | 6.1 [6.8–5.3] | - |
| | $E_{max}$ (R, %C) | 0.008 (100) | 0.014 (186)* | 0.007 (86) | 0.001 (13) |
| | $E_{max}$ (%) | 11 | 55* | 11 | - |
| **Fentanyl** | $pEC_{50}$ | 6.7 [7.1–6.2] | 7.0 [8.2–5.8] | 6.7 [7.2–6.3] | - |
| | $E_{max}$ (R, %C) | 0.04 (100) | 0.02 (47)* | 0.03 (75) | 0 (5)* |
| | $E_{max}$ (%) | 61 | 83 | 58 | - |

Concentration response curves from Fig. 2 were analysed using a three-parameter fit (Materials and Methods). Values represent mean [CI]. Emax is expressed as BRET ratio (R), % of control cells (%C) and % of DAMGO in corresponding cell line (%).
*$p < 0.05$ compared to control cells, unpaired T-test.

GRK5 overexpression was more pronounced for DAMGO and fentanyl. While GRK5 overexpression enhanced the recruitment induced by morphine, the effect was not as robust as when GRK2 or GRK3 were overexpressed (Fig. 3).

#### Dual role of Thr[376] and Thr[379] in promoting β-arrestin2 recruitment
The results above show that while phosphorylation of Thr[376] and Thr[379] by GRK2/3 is required for robust β-arrestin2 recruitment, overexpression of individual GRKs can rescue the compromised recruitment upon knock-out of all GRKs. We therefore hypothesised that GRK overexpression would not be sufficient to facilitate robust β-arrestin2 recruitment to a MOR mutant that, while retaining the essential Ser[375], cannot reach further higher order phosphorylation. To this end we generated a MOR harbouring Thr[376]Ala and Thr[379]Ala mutations (STANT-2A MOR). In control cells, expressing endogenous GRKs, the STANT-2A MOR mutant failed to recruit β-arrestin2 when activated by DAMGO, morphine or fentanyl (Fig. 4A). This effect was not due to differences in expression or function of the STANT-2A mutant compared to WT MOR, (Supp Fig. 2). Importantly, when STANT-

2A MOR was expressed in ΔQ-GRK cells together with GRK2 or GRK5, a limited rescue of β-arrestin2 recruitment was observed that did not reach the recruitment levels observed for the WT receptor upon overexpression of these two GRKs (Fig. 4B, C, Table 3). These data support our hypothesis that Thr[376] and Thr[379] are important for the facilitation of robust β-arrestin2 recruitment to the MOR.

To further investigate the engagement of GRKs with the MOR, we measured the recruitment of the different GRK isoforms to the MOR WT and STANT-2A mutant using BRET. In agreement with their plasma membrane localisation, the baseline BRET signal between MOR (WT and STANT-2A) and GRK5 or GRK6 was higher than that of GRK2 or GRK3 and resulted in the absence of a detectable ligand-induced response (Fig. 5A). Ligand-induced GRK2-YFP and GRK3-YFP recruitment to MOR WT was very similar to β-arrestin2 recruitment despite showing increased potencies and morphine relative efficacy, reflecting the known effect of GRK overexpression (Fig. 5B, C, Table 4). Interestingly, mutation of Thr[376] and Thr[379] to Ala, resulted in a significant decrease in the magnitude of GRK/MOR BRET response for all ligands (Fig. 5B, C, Table 4), suggesting that

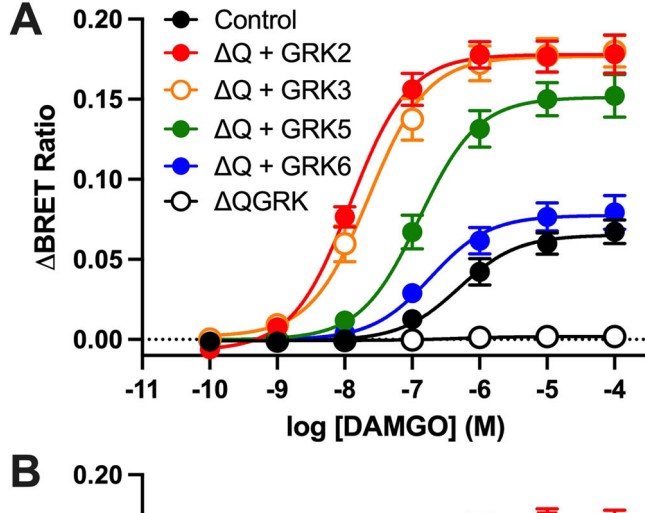

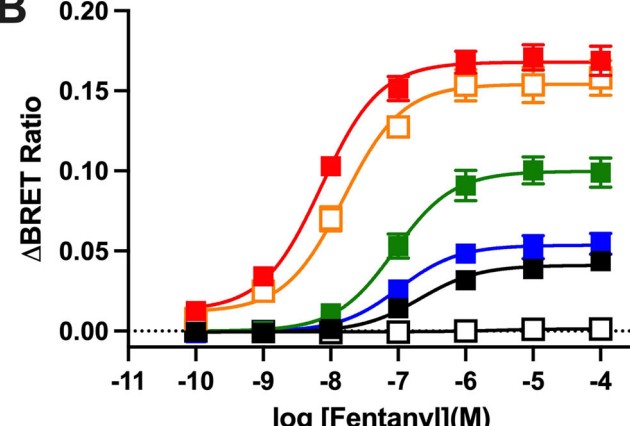

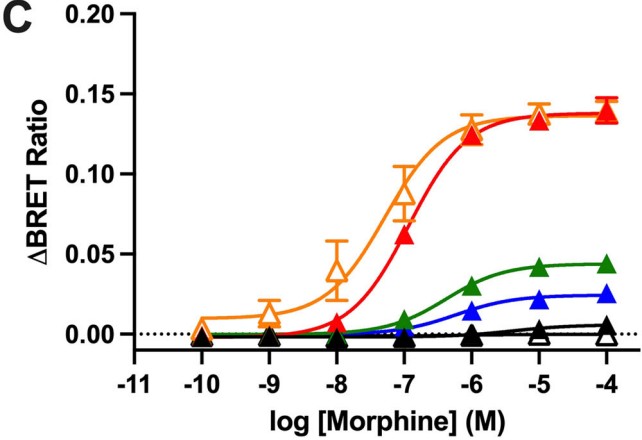

**Fig. 3 | β-arrestin2 recruitment to the MOR upon overexpression of a single GRK.** β-arrestin2-Venus recruitment to the Flag-mMOR-Nluc upon stimulation with DAMGO (**A**), fentanyl (**B**) or morphine (**C**) for 10 min at 37 °C in quadruple GRK knockout cells (ΔQ-GRK), overexpressing a single GRK (ΔQ + GRK) or expressing all endogenous GRKs (Control). Data show the mean ± SEM baseline-corrected BRET ratio of 5 independent experiments performed in triplicate.

these residues are not only key phosphorylation sites, but are also important for the recruitment of GRK2 and GRK3 to the receptor.

## Discussion

The role of GRKs, in particular GRK2, in facilitating MOR β-arrestin recruitment and internalisation was described upon the realisation that distinct MOR agonists induced internalisation and β-arrestin translocation to the plasma membrane to different extents, and that this could be altered

upon overexpression of GRKs[19,31]. As GRK2 knockout is embryonically lethal in mice[32], the role of this kinase in MOR function (and of GPCRs in general) has mainly been studied using pharmacological inhibitors[33] or knock-down strategies[8], which, while providing important molecular insights, are limited in their interpretation. Here we investigated the impact of GRK subfamily knock-out on ligand-dependent MOR phosphorylation and β-arrestin2 recruitment. We show that both GRK2/3 and GRK5/6 subfamilies promote phosphorylation of Thr[370] and Ser[375], however only GRK2/3 induce the higher-order phosphorylation of Thr[376] and Thr[379]. Moreover, we identify these two residues as key phosphor-sites with dual roles; participating in GRK2/3 recruitment and promoting robust β-arrestin2 recruitment. Given the requirement of β-arrestin2 recruitment to promote effective MOR internalisation, and the fact that opioid agonists can be differentiated based on their ability to induce higher order phosphorylation, these findings provide molecular insights to these established differences.

Using the MOR as a model system to validate the collection of Δ-GRK clones used here, Drube et al.[25] identified Thr[376] as a specific target of GRK2 and 3 and showed that Thr[370], Ser[375] and Thr[379] can be phosphorylated by all GRKs (albeit to a different extents) upon stimulation of the MOR with DAMGO. While these observations are mostly recapitulated here, we show that Thr[379] is also a specific target of GRK2/3 and demonstrate that morphine does not induce Thr[376] or Thr[379] phosphorylation under endogenous expression conditions, supporting its limited β-arrestin2 recruitment. In another study, Moller et al.[27] used GRK2 and/or GRK3 knock out cells to suggest a major role of GRK2 for MOR β-arrestin recruitment and internalisation that is supported by our results, although no link to receptor phosphorylation was described in this work.

The phosphorylation data obtained here upon GRK knockout provide further insight into the contribution of GRK isoform families to the ligand-dependent phosphorylation barcode at the MOR. GRK2/3 mediate DAMGO- and fentanyl-induced phosphorylation of Thr[376] and Thr[379] while GRK5/6 are dispensable for such higher-order phosphorylation induced by these ligands. Our results also support previous findings suggesting that the morphine-activated MOR is a good substrate for phosphorylation by GRK5[8,15,18], although they also suggest that GRK2/3 can be involved in morphine-induced phosphorylation of Ser[375]. As we have previously demonstrated that phosphorylation of Ser[375] precedes, and is essential for, higher-order phosphorylation of Thr[376] or Thr[379] [4,9], our results also suggest that this requirement is specific to GRK2/3 as when Ser[375] is phosphorylated by GRK5/6 in ΔGRK2/3 cells, the distal residues remain unphosphorylated, even upon stimulation of the cells with high efficacy agonists. Altogether, our data reveal that, the higher-order phosphorylation of Thr[376] and Thr[379] is required for robust β-arrestin2 recruitment as partial modification of the [370]TREHPSTANT[379] sequence by phosphorylation of only Thr[370] and Ser[375] (as induced by morphine in control cells, or by all ligands in ΔGRK2/3 cells) results in a significantly reduced (but still concentration-dependent) β-arrestin2 recruitment. Thus, we identify a mechanism whereby following activation by high efficacy agonists, phosphorylation of Ser[375] facilitates the necessary phosphorylation of Thr[376] and Thr[379], which are responsible for the robust β-arrestin2 recruitment required for the internalisation of this receptor. This explains how strategies that bypass the limited modification of the [370]TREHPSTANT[379] sequence by partial agonists and induce overall homogenous phosphorylation, such as overexpression of GRK2/3, will lead to dramatic increases in β-arrestin2 recruitment windows.

While it is tempting to speculate that GRK2/3 phosphorylation of Thr[370] and Ser[375] is required for further higher order phosphorylation of Thr[376] and Thr[379] by these kinases, it cannot be discounted that phosphorylation of Thr[370] and Ser[375] by any other kinase (e.g. GRK5/6) can facilitate the GRK2/3-mediated modification of Thr[376] and Thr[379]. Indeed, while CRISPR/Cas9 knock-out studies are useful to identify key signalling effectors, it must be considered that by eliminating the expression of a certain protein, the competition between mediators of the same effect is severely altered; namely, that if the main effector is knocked-out, the action of other, less efficient effectors may become more apparent. Thus, the absolute

**Table 2 | Potency (pEC$_{50}$) and maximal effect (E$_{max}$) for DAMGO, morphine and fentanyl β-arrestin2 recruitment in control and GRK knock-out cells**

|  |  | Control | ΔQ + GRK2 | ΔQ + GRK3 | ΔQ + GRK5 | ΔQ + GRK6 | QGRK |
|---|---|---|---|---|---|---|---|
| **DAMGO** | pEC$_{50}$ | 6.3 [6.7–6] | 7.9 [8.1–7.7]* | 7.6 [7.9–7.4]* | 6.9 [7.1–6.7] | 6.7 [7.1–6.4] | - |
|  | E$_{max}$ (R, %C) | 0.07 (100) | 0.18 (280)* | 0.18 (265)* | 0.15 (229)* | 0.08 (118) | 0 (4)* |
|  | E$_{max}$ (%) | 100 | 100 | 100 | 100 | 100 | - |
| **Morphine** | pEC$_{50}$ | 5.4 [6.3–4.5] | 6.9 [7.0–6.8]* | 7.3 [7.7–6.9]* | 6.4 [6.7–6.1]* | 6.2 [6.5–6.0]* | - |
|  | E$_{max}$ (R, %C) | 0.008 (100) | 0.140 (1845)* | 0.127 (1669)* | 0.044 (581)* | 0.025 (324) | 0.001 (13)* |
|  | E$_{max}$ (%) | 11 | 78 | 70 | 29 | 32 | - |
| **Fentanyl** | pEC$_{50}$ | 6.7 [7.1–6.2] | 8.1 [8.3–8.0]* | 7.8 [8.0–7.5]* | 7.1 [7.3–6.8] | 7.0 [7.3–6.6] | - |
|  | E$_{max}$ (R, %C) | 0.04 (100) | 0.155 (373)* | 0.142 (342)* | 0.100 (240)* | 0.05 (130) | 0 (5)* |
|  | E$_{max}$ (%) | 61 | 94 | 79 | 65 | 68 | - |

Concentration response curves from Fig. 3 were analysed using a three-parameter fit (Materials and Methods). Values represent mean [CI]. Emax is expressed as BRET ratio (R), % of control cells (%C) and % of DAMGO in corresponding cell line (%).

*$p < 0.05$ compared to control cells, unpaired T-test.

**Fig. 4 | β-arrestin2 recruitment to the STANT-2A MOR.** β-arrestin2-Venus recruitment to the WT or STANT-2A MOR (where Thr[376] and Thr[379] were mutated to Ala) upon stimulation with DAMGO, fentanyl or morphine for 10 min at 37 °C in cells expressing endogenous GRKs (**A**) or in quadruple GRK knockout cells (ΔQ GRK), overexpressing GRK2 (**B**) or GRK5 (**C**). Data show the mean ± SEM baseline-corrected BRET ratio of 3 independent experiments performed in triplicate.

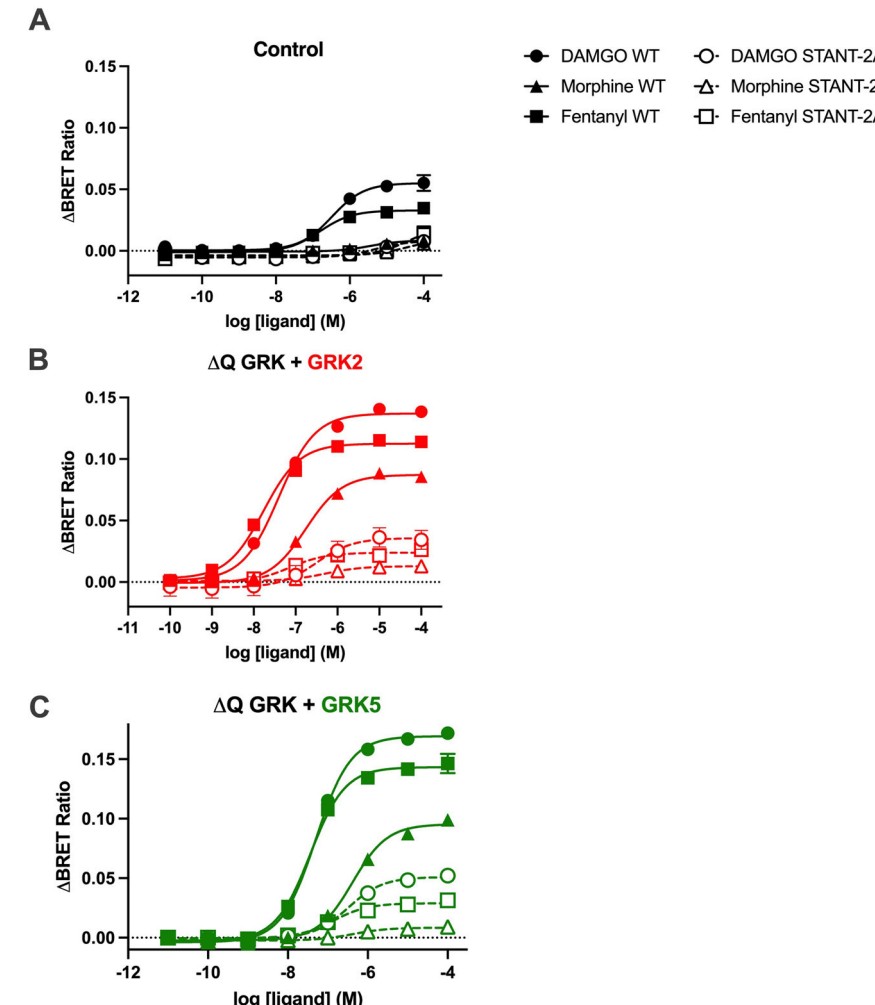

contribution of GRK5/6 vs GRK2/3 in phosphorylation of Thr[370] and Ser[375] remains to be elucidated. This is illustrated by the observation that β-arrestin2 recruitment by morphine is increased in ΔGRK2/3 cells, which suggests that in the absence of GRK2/3, other kinases (e.g GRK5/6, PKC) may have less competition and their effects become more apparent. In this context, PKC is an obvious candidate, as it has been shown to modify MOR in a morphine-dependent manner, and different than GRK2[34,35].

Re-introduction of the individual GRKs into ΔQ-GRK cells showed that each GRK isoform enhanced the MOR-β-arrestin recruitment to levels similar (GRK6) or higher (GRK2, 3 and 5) than induced by the combined endogenous expression of GRKs in control cells. We have previously shown that upon overexpression in ΔQ-GRK cells (using the same constructs and the same cells), the relative expression compared to endogenous (control) levels varies between ~4-fold (GRK2), ~8-fold (GRK3 and GRK6) and ~15-

**Table 3 | Potency (pEC$_{50}$) and maximal effect (E$_{max}$) for DAMGO, morphine and fentanyl β-arrestin2 recruitment in control and GRK knock-out cells expressing WT and STANT-2A MOR**

|  |  | Control WT | Control STANT-2A | ΔQ + GRK2 WT | ΔQ + GRK2 STANT-2A | ΔQ + GRK5 WT | ΔQ + GRK5 STANT-2A |
|---|---|---|---|---|---|---|---|
| **DAMGO** | pEC$_{50}$ | 6.5 [6.7–6.3] | - | 7.4 [7.6–7.2] | 6.5 [6.7–6.3] | 7.3 [7.4–7.2] | 6.5 [6.7–6.3] |
|  | E$_{max}$ (R, %C) | 0.06 (100) | - | 0.14 (175)* | 0.04 (-) | 0.17 (283)* | 0.05 (-) |
|  | E$_{max}$ (%) | 100 | - | 100 | 100 | 100 | 100 |
| **Morphine** | pEC$_{50}$ | 5.4 [7.3–3.1] | - | 6.8 [6.9–6.6] | 6.3 [6.5–6.1] | 6.4 [6.5–6.3] | 6.4 [7.5–5.2] |
|  | E$_{max}$ (R, %C) | 0.009 (100) | - | 0.09 (1000)* | 0.01 (-) | 0.10 (1000)* | 0.01 (-) |
|  | E$_{max}$ (%) | 29 | - | 64 | 25 | 59 | 20 |
| **Fentanyl** | pEC$_{50}$ | 6.8 [7.0–6.6] | - | 7.8 [8.0–7.5] | 7.1 [7.5–6.8] | 7.4 [7.6–7.3] | 6.9 [7.2–6.4] |
|  | E$_{max}$ (R, %C) | 0.03 (100) | - | 0.110 (342)* | 0.02 (-) | 0.15 (500)* | 0.03 (-) |
|  | E$_{max}$ (%) | 60 | - | 79 | 50 | 88 | 60 |

Concentration response curves from Fig. 4 were analysed using a three-parameter fit (Materials and Methods). Values represent mean [CI]. Emax is expressed as BRET ratio (R), % of control cells (%C) and % of DAMGO in corresponding cell line (%).
*$p < 0.05$ compared to control cells, unpaired T-test.

**Fig. 5 | GRK recruitment BRET to MOR.** GRK2/3/5/6-YFP recruitment to the WT or STANT-2A MOR (where Thr$^{376}$ and Thr$^{379}$ were mutated to Ala) upon stimulation with DAMGO, fentanyl or morphine for 10 min at 37 °C. **A** Raw BRET ratio between MOR (WT, top panel, STANT-2A lower panel) and GRKs upon vehicle or DAMGO, Fentanyl or morphine (10 μM) stimulation for 10 min. *$p < 0.05$ compared to vehicle, unpaired T-test. Concentration response curve for GRK2-YFP (**B**) or GRK3-YFP (**C**) recruitment to MOR. Data show the mean ± SEM baseline-corrected BRET ratio of 3 independent experiments performed in triplicate.

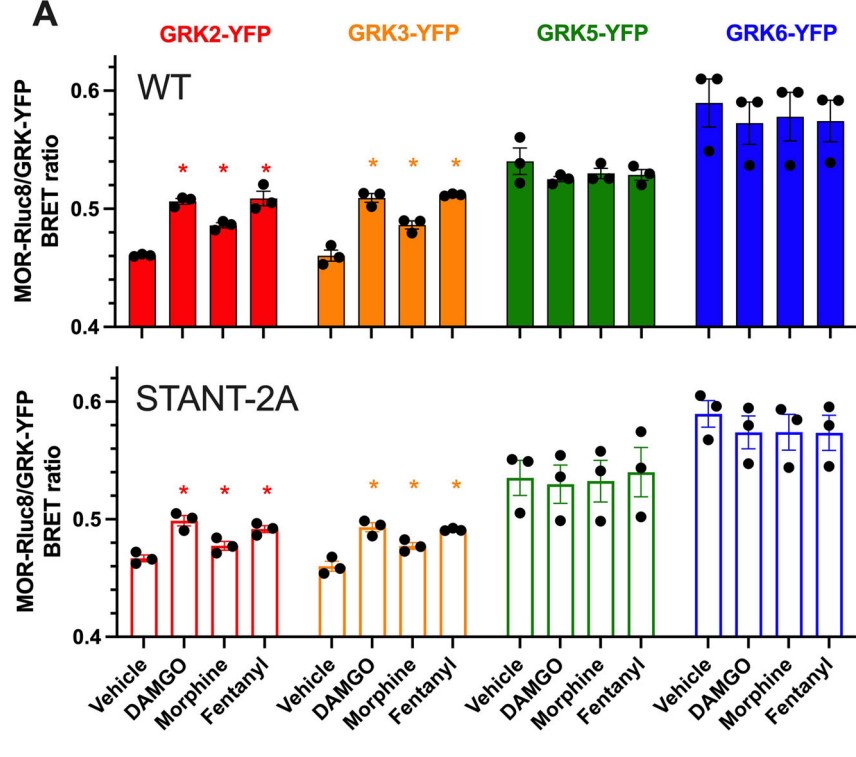

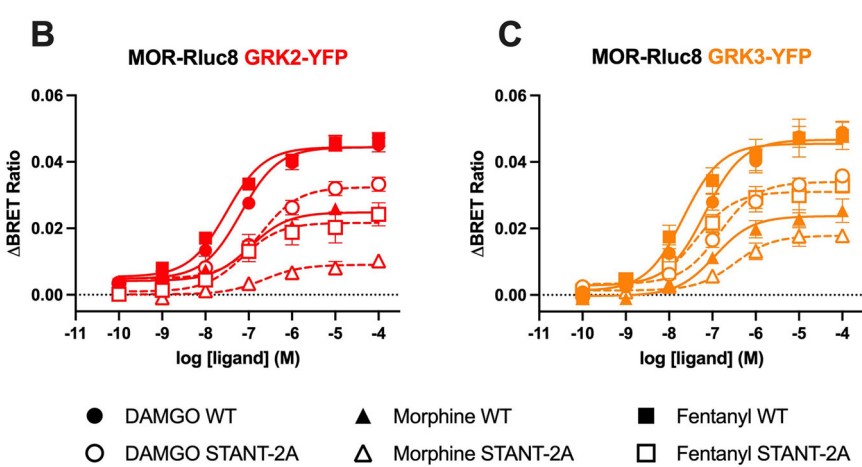

**Table 4 | Potency (pEC$_{50}$) and maximal effect (E$_{max}$) for DAMGO, morphine and fentanyl GRK2 and GRK3 recruitment in cells expressing WT and STANT-2A MOR**

|  |  | GRK2 WT | GRK2 STANT-2A | GRK3 WT | GRK3 STANT-2A |
|---|---|---|---|---|---|
| **DAMGO** | pEC$_{50}$ | 7.2 [7.0–7.4] | 6.7 [5.9–7.6] | 7.2 [7.0–7.3] | 6.8 [6.0–7.5] |
|  | E$_{max}$ (%) | 0.039 (100) | 0.028 (100)* | 0.044 (100) | 0.031 (100)* |
| **Morphine** | pEC$_{50}$ | 7.0 [6.2–7.6] | 6.7 [6.5–6.8] | 6.9 [6.9–7.0] | 6.5 [6.1–6.8]* |
|  | E$_{max}$ (%) | 0.021 (54) | 0.009 (32) | 0.024 (54) | 0.014 (45)* |
| **Fentanyl** | pEC$_{50}$ | 7.5 [7.3–7.7] | 7.1 [6.7–7.5]* | 7.7 [7.3–8.0] | 7.4 [7.2–7.5]* |
|  | E$_{max}$ (%) | 0.039 (100) | 0.020 (74) | 0.044 (100) | 0.028 (68)* |

Concentration response curves from Fig. 5 were analysed using a three-parameter fit (Materials and Methods). Values represent mean [CI]. Emax is expressed as BRET ratio and % of DAMGO in corresponding cell line.
*$p < 0.05$ compared to WT, unpaired $t$-test.

fold (GRK5)[25]. Thus, it is unlikely that the differential effects observed upon GRK2 and GRK3 overexpression are due to higher expression compared to GRK5 and GRK6. Altogether these data suggest that ultimately, the relative tissue expression of GRK isoforms defines their specific contributions to receptor phosphorylation and ensuing events that require this modification[12].

Our results show that GRKs mediate the phosphorylation of the $^{370}$TREHPSTANT$^{379}$ motif and that such phosphorylation is required for β-arrestin2 recruitment as in ΔQ-GRK cells both events are abolished. However, it is known that other kinases can also mediate phosphorylation of other residues within the C-tail of MOR. For example, PKC can phosphorylate Ser$^{363}$ [36,37] a site that was shown to retain strong phosphorylation in ΔQ-GRK cells[25]. PKC is involved in morphine-induced signalling[38,39], desensitization[40] and tolerance[16]. Ser$^{363}$ has been shown to be the primary substrate for PKC-mediated phosphorylation with mutation of this residue to alanine impacting heterologous receptor desensitization[36,37]. However, as with GRKs, most reports to date have used pharmacological inhibitors or knock-down approaches. With the recent generation of typical and atypical PKC isoform CRISPR/Cas9 knock out cells[41], the relative contribution to MOR signalling of GRK and PKC isoforms upon morphine stimulation can be further investigated in the future.

Finally, in the context of the MOR and separately from their key role in β-arrestin recruitment and MOR internalisation, GRKs have been described to participate in other receptor-mediated functions. For example, we have previously shown that GRK2 is required for agonist-induced changes in MOR diffusion across the plasma membrane[42]. Using dual-colour single molecule tracking, the AT1R and GRK2/5 molecules have been observed to be concentrated in a similar domain and it has been suggested that such accumulation in so called "hot spots" is the molecular basis for efficient receptor phosphorylation[26]. This raises the possibility that a similar mechanism may underlie the robust phosphorylation induced by high efficacy MOR agonists whereby MOR and GRK2 concentrate in specific membrane domains, facilitating higher-order phosphorylation. Future studies should investigate the existence of ligand-induced MOR/GRK "hot-spots" as well as the diffusion of MOR upon knock-out of specific GRK isoforms.

The data obtained here also aligns with recent classifications of GPCRs in terms of their GRK engagement and phosphorylation patterns. Using a library of synthetic phosphopeptide analogues of the C-terminus of rhodopsin and nuclear magnetic resonance (NMR), different functional classes of phosphorylation sites within GPCRs C-tails have been identified[43]. Two "key sites" are required for arrestin binding and activation, an "inhibitory site" negatively impacts arrestin binding, and "modulator sites" influence the global conformation of arrestin. Interestingly, upon sequence alignment of receptor C-termini from several GPCRs, the Thr$^{376}$ and Thr$^{379}$ residues identified here, are suggested to be the "key sites" for the MOR[43], supporting our hypothesis that these specific sites are dictating the strength of β-arrestin2 binding to the MOR upon binding to high efficacy agonists. According to this model, the MOR would lack the inhibitory site as there is no negatively charged residue between the two key sites, while Thr$^{370}$ and

Ser$^{375}$ align with the modulator sites, which can be phosphorylated by agonists of low and high efficacy.

While structural information regarding the interactions of β-arrestins (mostly β-arrestin 1) with GPCRs is starting to emerge[44], to-date, there is only two cryo-EM structures of a GPCR in complex with GRKs, namely the rhodopsin-GRK1 and the NTS1R-GRK2 complexes[45,46]. In all these structures, it is readily apparent that the arrestin and GRK complexes exhibit high conformational heterogeneity, which is likely a consequence of their characteristic ability to adapt and bind to hundreds of GPCRs. The recent cryo-EM structures of Rho*-GRK1 and NTSR1-GRK2 demonstrate that the N-terminal end of the αN helix, highly conserved in all GRKs, directly inserts within the cyto-plasmic cleft of the activated receptor. The Rho*-GRK1 structure suggests interactions of ICL1 loop and H8 of Rho* with GRK1. Interestingly, the basic residues in ICL1 and Arg$^{8.51}$ in H8, are highly conserved in class A GPCRs, including the MOR. Based on the NTSR1–GRK2 complex structures, the extended loop of ICL3 or the elongated C-terminal tail of the GPCR can reach the active cleft of GRK2 and thus be available for phosphorylation by this kinase (in contrast to ICL1 and ICL2, unlikely to be accessible to the kinase active site). Given the data presented here, we can only speculate that the availability of more distal sites of the C-tail (including Thr$^{376}$ and Thr$^{379}$ of the MOR) to the active site of GRK5/6 is more limited, explaining their differential phosphorylation pattern when endogenously expressed. Indeed, increasing evidence is starting to sug-gest that GRK5/6 tend to phosphorylate sites "proximal" to the plasma membrane within the C-terminus of a GPCR, while GRK2/3 are able to phosphorylate more "distal" sites[47]. Our data agrees with this "geometric" model, with Thr$^{370}$ and Ser$^{375}$ being "proximal" while Thr$^{376}$ and Thr$^{379}$ would be the "distal", GRK2/3 sites. It is tempting to speculate that such distinction is related to the tethering of GRK5/6 to the plasma membrane and the relative receptor-GRK complex geometry. However, despite the recent structure determination of GPCR-GRK complexes[45,46], experiments to test this hypothesis remain to be conducted.

In summary, our results identify GRK2/3-mediated phosphorylation of key residues Thr$^{376}$ and Thr$^{379}$ as a requirement for robust β-arrestin2 recruitment at the MOR by high efficacy agonists. This higher order phosphorylation requires a phosphorylated Ser$^{375}$ and can be overcome by overexpression of GRKs. These data are important in the context of the differential tissue expression of GRK isoforms[12], and will help under-standing the actions of different opioid agonists in different tissues, in particular in terms of MOR regulation.

## Materials and Methods
### Materials
[D-Ala2, N-Me-Phe4, Gly5-ol]-Enkephalin acetate salt (DAMGO) was purchased from Sigma Aldrich (Munich, Germany), morphine sulphate was purchased from Hameln Inc. (Hameln, Germany) and fentanyl citrate was purchased from Rotexmedica (Trittau, Germany). Polyethylenimine (PEI), Linear (MW 25,000) was purchased from Polysciences, Inc. Fur-imazine was from Promega.

PierceTM HA epitope tag antibody was obtained from Thermo Scientific (Rockford, IL, USA). The rabbit polyclonal phosphosite-specific μ-opioid receptor antibodies anti-pT370 (7TM0319B), anti-pT376 (7TM0319D), anti-pT379 (7TM0319E), anti-pS375 (7TM0319C) and anti-HA antibody (7TM000HA) were obtained from 7TM Antibodies (Jena, Germany)[47]. The secondary horseradish peroxidase (HRP)-linked anti-rabbit antibody was purchased from Cell Signaling (Frankfurt, Germany).

### cDNA constructs

Flag-mMOR-NLuc, β-arrestin2-Venus and human GRK2/3/5/6 constructs and their -YFP fusions have been previously described[4,20,25]. mGsi-Venus was from N. Lambert (Augusta University, Augusta, Georgia, USA). mMOR STANT-2A was generated by site directed mutagenesis following manufacturer's instructions.

### Cell culture and transfection

Control, ΔGRK 2/3, ΔGRK5/6, ΔQ-GRK HEK293 cells have been previously described[25]. Cells were grown in Dulbecco's Modified Eagle's Medium (DMEM) (Sigma Aldrich) supplemented with 10% v/v FBS (Sigma Aldrich) at 37 °C in a humidified incubator with 5% $CO_2$. For transfection, cells were seeded in 10-cm$^2$ cell culture dishes and transfected with 4 μg β-arrestin2-Venus, mGsi-Venus or GRK-YFP and 1 μg Flag-MOR-NLuc (or STANT-2A mutant) using a 1:6 DNA:PEI ratio. In GRK overexpression experiments, cells were additionally transfected with 2 μg of the corresponding GRK isoform.

### Western Blot analysis

HEK293 cells (control, ΔGRK 2/3, ΔGRK5/6, ΔQ-GRK) stably expressing HA-mMOR were seeded onto poly-L-lysine-coated 60 mm dishes and grown to 90% confluency. After 30 min agonist stimulation (DAMGO $10^{-5}$ M, morphine $10^{-5}$ M, fentanyl $10^{-6}$ M) at 37 °C, cells were washed and lysed in RIPA buffer (50 mM Tris-HCl, pH 7.4, 150 mM NaCl, 5 mM EDTA, 1% Nonidet P-40, 0.5% sodium deoxycholate, 0.1% SDS) containing protease and phosphatase inhibitors (Complete mini and PhosSTOP; Roche Diagnostics, Mannheim, Germany). Pierce$^{TM}$ HA epitope tag antibody beads (Thermo Scientific, Rockford, IL, USA) were used to enrich HA-tagged MOR following the manufacturer's instructions. The beads were then washed three times with RIPA buffer containing protease and phosphatase inhibitors. To elute proteins from the beads, the samples were incubated in SDS sample buffer (125 mM Tris pH 6.8, 4% SDS, 10% glycerol, 167 mm DTT) for 25 min at 43 °C. Supernatants were separated from the beads, loaded onto 8% SDS polyacrylamide gels, and then immunoblotted onto nitrocellulose membranes. After blocking (5% milk in TBS-T), membranes were incubated with anti-pT370 (7TM0319B), antipS375 (7TM0319C), anti-pT376 (7TM0319D), or anti-pT379 (7TM0319E) antibody overnight at 4 °C (7TM Antibodies, Jena, Germany). Membranes were incubated in HRP-linked secondary antibody for 2 h prior to detection using a chemiluminescence system (90 mM p-coumaricacid, 250 mM luminol, 30% hydrogen peroxide). Blots were subsequently stripped and re-incubated with the phosphorylation-independent anti-HA antibody to confirm equal loading of the gels. Note that similar expression of HA-MOR was also validated using a based-bead assay (Supp Fig. 5).

### β-arrestin2 and GRK recruitment BRET

24 h after transfection cells were harvested and transferred into white 96-well Poly-D-Lysine coated CulturPlates (PerkinElmer) in DMEM + 10% FBS. 48 h post transfection, the media in each well was aspirated, washed with Hank's balanced salt solution pH 7.4 (HBSS), replaced with HBSS and then kept at 37 °C for the remainder of the assay. Cells were then treated with 10 μL of ligand at 10x final concentration and incubated for 5 min, followed by addition of 10 μL furimazine

(final concentration 5 μM) and incubated for 5 min. Plates were then read on a PHERAstar Plate Reader (BMG LABTECH, Ortenberg, Germany) using the BRET1 filter set (535 ± 30 nm(fluorescence), 475 ± 30 nm luminescence). Raw BRET signals were calculated as the emission intensity at 520–545 nm divided by the emission intensity at 475–495 nm, and the vehicle-subtracted BRET ratio (drug-induced increase in BRET) was calculated and plotted.

### Data analysis

The results of concentration response experiments were analysed using Prism 10 (GraphPad Software Inc., San Diego, CA) and fitted using the following three parameter equation:

$$response = bottom + \frac{top - bottom}{1 + 10^{(\log EC_{50} - \log[A])}} \qquad (1)$$

Where top and bottom represent the maximal and minimal asymptote of the concentration response curve, [A] is the molar concentration of agonist, and $EC_{50}$ is the molar concentration of agonist required to give a response half-way between maximal and minimal asymptote.

### Statistics and reproducibility

Data show the mean ± standard error of the mean (SEM) from at least three separate experiments. The number of experimental repeats is stated in the corresponding Figure legends. Statistical analyses were performed using GraphPad Prism 10 software (San Diego, CA, USA). Statistical significance of data was tested using either unpaired, two-tailed t-test or one-way ANOVA. Throughout the study, $P < 0.05$ was used as the level of significance. Normal distribution of the data was assessed using the Shapiro-Wilk test in GraphPad Prism.

Graphical Abstract and Fig 1A were created using BioRender.com.

### Reporting summary

Further information on research design is available in the Nature Portfolio Reporting Summary linked to this article.

## Data availability

All data supporting the findings of this study are available within the article and its supplementary information files (uncropped blots in Supplementary Fig. 4 and phosphorylation statistical analysis in Supplementary Data 1). Additional information, relevant data and unique biological materials will be available from the corresponding author upon reasonable request. Source data are provided with this paper.

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

## Acknowledgements

This work was funded by the Academy of Medical Sciences (M.C.), the BBRSC (M.C, BB/T013966/1). O.U. was funded by MRC IMPACT

Programme. The authors would like to thank Ulrike Schiemenz for expert technical assistance.

## Author contributions
O.U. performed and analysed all BRET assays with assistance of J.G., S.F. performed and analysed all phosphorylation assays, NKB measured MOR expression levels, J.D. generated all GRK knockout cells, A.G. generated MOR mutants. S.S. provided phosphosite specific antibodies and supervised phosphorylation assays. M.C. developed and supervised the project with input from S.J.B, S.S. and C.H.; M.C. wrote the manuscript; all other authors revised the manuscript and gave final approval.

## Competing interests
S.S. is the founder and scientific advisor of 7TM Antibodies GmbH, Jena, Germany. All other authors declare no competing interests.
