## [Transparent Peer Review file · Communications Biology]

Key phosphorylation sites for robust β -arrestin2 binding at the MOR revisited

Corresponding Author: Professor Meritxell Canals

Figures originally included in the author's rebuttal have been redacted from this file.

Version 0:

Reviewer comments:

Reviewer #1

(Remarks to the Author)

The manuscript by Underwood and colleagues investigates 1) MOR phosphorylation at distinct residues and 2) b-arrestin2 recruitment to MOR in response to the agonists DAMGO, fentanyl, and morphine. The goal of the study is to investigate the contribution of the kinases GRK2/3 and GRK5/6 to these two regulatory mechanisms downstream of MOR activation. For this, the authors employ GRK OK cells, previously published by Carsten Hoffmann's group.

Specific / major points:

- As accurately cited and discussed by the authors in the introduction and discussion sections, a large body of published work has already established the ligand-dependent effects on MOR phosphorylation and b-arrestin binding and the role of GRK2/3 and GRK5/6. The presented line of research is a recapitulation of existing data, and the novelty of the results is not clear.
- Can the authors go beyond existing findings and investigate the agonist selective recruitment of GRK subtypes to the MOR? Are GRK2/3 vs GRK 5/6 able to sense agonist specific receptor conformations? What is the engagement mode of GRK2/3 vs GRK 5/6 with the MOR? Any results that would address these questions would add novelty. As one inroad, the authors could measure the recruitment of GRK2/3 and GRK5/6 to MOR and MOR mutants (such as STANT-2A) upon activation with different ligands.
- No statistical significance test has been performed on any data shown in the figures. If the authors want to establish that GRK2/3 or GRK5/6 play (or not) a role in the phosphorylation of a certain MOR residue, they need to provide accurate significance testing (indicating the test used and the number of independent experiments). For example, Fig. 1C needs to provide information if there is a significant difference between phosphorylation states in the control and KO cell lines. This is particularly unclear for pS375 phosphorylation after morphine treatment, where the phosphorylation levels seem similar between GRK2/3 KO and GRK5/6 KO cells, despite the authors' conclusion that GRK5/6 (rather than GRK2/3) mediate pS375 phosphorylation.
- Data in Suppl. Fig. 1 shows ligand dependent G protein (mGsi) engagement with MOR in different GRK KO cell lines relative to control cells. The authors conclude 'the ability of MOR to interact with G proteins, as measured by recruitment of mGsi, was not affected in any of the knock-out cells'. Yet no statistical testing is performed (e.g. for Emax) and contrary to the authors' comment, it appears that mGsi recruitment in control vs KO cells (e.g. S1A vs S1C) is strongly reduced.
- The authors present data on rescued / enhanced b-arrestin2 recruitment upon reintroducing individual GRK family members into quadruple GRK2/3/5/6 KO cells. They draw conclusions on the ability of different GRKs to rescue / increase the b-arrestin2 engagement relative to control cells. To make this conclusion, the authors need to probe the level of GRK over-expression across all conditions. Are GRK2/3 more strongly expressed than GRK5/6? And how does the over-expression level compare to the endogenous GRK level of control cells?
- The authors need to accurately describe what the datapoints in the figures illustrate. For example, Suppl. Fig. 2A does not specify what the presented data points are, Suppl. Fig. 2B indicates '3-10 independent experiments' but shows exactly three datapoints for both conditions, Suppl. Fig. 3 indicates 3-10 independent experiments, yet the graph clearly plots more data points for each condition than 3-10.

- The authors need to provide more information on the plasmids / cDNA constructs used (in addition to citing the source papers). Are the authors working with receptor, arrestin, GRK constructs of human/mouse/canine/... origin? Also, how much DNA (receptor, arrestin, and GRK plasmids) is transfected in the b-arrestin2 recruitment assays?

Reviewer #2

(Remarks to the Author)

In this manuscript, Underwood et al. identified the roles of different GRK sub-families that can promote unique phosphorylation patterns on mu opioid receptor (MOR). The approach was straightforward by using knockout cell lines and rescue strategies. The conclusion could be more convincing, and the work be more useful by addressing the following points.

Major comments:

1. The authors didn't explain the interesting observation that over-expression of single GRK could rescue and even enhance the Barr2 recruitment in the GRK knock-out cells. This could happen, possibly caused by different expression levels of GRKs, elevated levels of the phosphorylated MOR, changes in the phosphorylated barcode, or other potential factors?
2. Based on the first comment, it is reasonable to speculate that the creation of distinct phosphorylation barcodes by individual endogenous GRKs may be attributed to variations in the expression levels of GRKs. Therefore, it is important to address the first comment and quantify the expression levels of endogenous GRKs.
3. Although the authors identified GRK2/3 could phosphorylate all four sites and GRK5/6 only two sites, it is still unclear whether individual GRK promotes unique phosphorylation codes? For example, Do GRK2 and GRK3 target the same site? GRK5 and GRK6 on the same site as well? GRK5 and GRK6 appear to behave differently based on Figure 3. One experiment they could do is in the rescue experiments after introducing individual GRKs, they can determine the GRK-specific phosphorylation site.
4. It would be very useful to add beta-arrestin1 in the same functional test as beta-arrestin2 for comparison.
5. In line 192, why do morphine in GRK2/3 knockout cells recruit beta-arrestin2 5-fold higher than the wt cells?

Minor points:

6. While authors have already done comprehensive statistical analysis throughout the manuscript, there are still some areas lacking this analysis, such as Figure 1C. For Table 2, the authors claimed that statistical analysis has been performed, but there are no corresponding labels provided.
7. Many GPCRs have phosphorylation sites in the intracellular loop regions. There are a few Ser/Thr residues located in the ICLs of MOR. The authors didn't look at them or is there no contribution from the ICL regions? This could be discussed.

Reviewer #3

(Remarks to the Author)

The manuscript "Key phosphorylation sites for robust b-arrestin2 binding at the MOR revisited" uses an innovative set of experiments with CRISPR-KO cell lines, phosphosite-antibodies and site-directed mutagenesis to clearly show phosphorylation patterns of 3 mu opioid receptor (MOR) agonists involved in the recruitment of b-arrestin. Importantly the studies compare phosphorylation patterns with endogenous levels of GRK 2,3,5,6 with complete KO of all GRKs, as well as with over-expression of each single GRK on the complete KO background. The manuscript is very clearly written with appropriate attention to previous studies and their caveats. The data presented are based on appropriate numbers of replicates with appropriate controls. Overall, the manuscript is an important contribution to the field and indicates that tissue expression and localization of GRK complements will play a critical role in determining the desensitization, internalization and trafficking of MORs. Although other studies have proposed this, the studies in this manuscript for the first time clearly link GRK-mediated phosphorylation patterns to ligand-dependent differences in MOR recruitment of b-arrestin2.

The only confusing statement in the manuscript is the sentence (lines 193-195) that recruitment of mGsi was not affected in any of the knock-out cells (Supp Fig 1). The figure shows differences in recruitment but there are no statistical analyses, thus it is not clear what the basis of the statement is.

Reviewer #4

(Remarks to the Author)

In this study, Underwood and collaborators aimed to identify key phosphorylation sites in the mu opioid receptor (MOR) involved in beta-arrestin2 recruitment upon some opioid agonist. All the study was realized in a heterologous system (HEK293 cells transfected with different constructs). Using HEK293 cell lines knocked down for GRKs, phosphosite specific antibodies and site-directed mutagenesis, they demonstrated that the GRK subfamilies (GRK2/3 or GRK5/6) differentially contribute to ligand-induced phosphorylation barcodes and -arrestin2 recruitment. The paper is very well written and the experiments were well conducted and suitable to answer the scientific question.

However, I have some remarks and questions:

- In the abstract: I suggest removing the term of "desensitisation" as no data on desensitisation are provided in the study.

- In the material and methods :
- In the HA-MOR construct, is it mouse MOR ? I'm not sure it is specified.
- The authors used different MOR constructs (HA-MOR, Flag-mMOR-NLuc). But, did they check that the ligands used in the study bind and activate these different MOR constructs in the same manner ? It is important to link phosphorylation studies and beta-arrestin2 recruitment studies.
- Regarding the statistical analysis, the authors mentioned "Statistical significance was determined using an unpaired T-Test, corrected for multiple comparisons using a Holm-Šidák test (*: $p \leq 0.05$).” I am a bit confused here. Indeed, T-test are used to compared two groups and the Holm-Šidák correction for multiple test is used after an ANOVA, used to compare more than 2 groups. So, this must be clarified by mentioning which statistical analysis was used in each experiment and the subsequent results. For instance, in the case of ANOVA, F statistic should be given along with results of the multiple comparisons. Moreover, did the authors verify the normality of the distribution as well as the homoscedasticity of the variance ? If yes, which test did they use ? Finally, it is mentioned that " Data show the mean \pm SEM of at least 5 independent experiments performed in triplicate” but in some figures, it is mentioned "3 independent experiments".
- For the phosphorylation experiments, how the authors choose the agonist concentrations ?
- In figure 1:
 - Fig 1B: the np-MOR bands look more intense in control cells. Did the authors quantify receptor expression in these different cell lines ?
 - Fig 1C: I'm bit surprised that no SEM appeared on DAMGO groups. Even if it was normalized to DAMGO, SEM should appeared. Why no statistical analysis was performed ?
- In figure 3 and table 2 : The recruitment of beta-arrestin2 appears to vary depending on whether GRK2 or GRK3 is overexpressed. How could it be explained ?

Author Rebuttal letter:

Reviewers' comments:

Reviewer #1 (Remarks to the Author):

The manuscript by Underwood and colleagues investigates 1) MOR phosphorylation at distinct residues and 2) b-arrestin2 recruitment to MOR in response to the agonists DAMGO, fentanyl, and morphine. The goal of the study is to investigate the contribution of the kinases GRK2/3 and GRK5/6 to these two regulatory mechanisms downstream of MOR activation. For this, the authors employ GRK OK cells, previously published by Carsten Hoffmann's group.

Specific / major points:

- As accurately cited and discussed by the authors in the introduction and discussion sections, a large body of published work has already established the ligand-dependent effects on MOR phosphorylation and b-arrestin binding and the role of GRK2/3 and GRK5/6. The presented line of research is a recapitulation of existing data, and the novelty of the results is not clear.

We respectfully disagree with the statement that "the presented line of research is a recapitulation of existing data". While there is, indeed, a large body of work investigating the role of MOR phosphorylation in receptor regulation, our work is original and novel in showing the clear link of specific differences in GRK-mediated phosphorylation patterns with ligand-dependent differences in MOR recruitment of μ -arrestin2 (as highlighted by Reviewer 3).

Given the extensive literature on this subject, we have tried to highlight previous studies and their caveats, while providing new insight into previously unappreciated determinants of ligand- and context-dependent differences in MOR regulation by GRKs.

Specifically, the following findings have not been reported previously. First, both GRK2/3 and GRK5/6 subfamilies promote phosphorylation of Thr370 and Ser375, however only GRK2/3 induce the higher-order phosphorylation of Thr376 and Thr379. We have achieved this through selective GRK KO, previous pharmacological inhibitors and genetic strategies (si/shRNAs) have not been able to directly assess the contribution of specific kinase isoforms to the ligand-induced MOR phosphorylation barcode. Second, Thr376 and Thr379 are key phosphorylation sites essential to promote robust μ -arrestin2 recruitment. This concept evolves from the current understanding, focused mostly on the role of Ser375 as the driver of μ -arrestin recruitment to the receptor. Thus, we identify a mechanism whereby following activation by high efficacy agonists, phosphorylation of Ser375 facilitates the necessary phosphorylation of Thr376 and Thr379, which is responsible for the robust μ -arrestin2 recruitment required for the internalisation of this receptor. This explains how strategies that bypass the limited modification of the 370TREHPSTANT379 sequence by partial agonists and induce overall homogenous phosphorylation, such as overexpression of GRK2/3, will lead to dramatic increases in μ -arrestin2 recruitment windows by this family of ligands. Finally, our work shows that ligand-dependent differences in arrestin recruitment are due to their ability

to differentially induce phosphorylation of Thr376 and Thr379 whereas previous work highlighted the importance of the Ser375 and the 370TREHPSTANT379 motif as a whole.

- Can the authors go beyond existing findings and investigate the agonist selective recruitment of GRK subtypes to the MOR? Are GRK2/3 vs GRK 5/6 able to sense agonist specific receptor conformations? What is the engagement mode of GRK2/3 vs GRK 5/6 with the MOR? Any results that would address these questions would add novelty. As one inroad, the authors could measure the recruitment of GRK2/3 and GRK5/6 to MOR and MOR mutants (such as STANT-2A) upon activation with different ligands.

This is an insightful suggestion from the Reviewer. We have now assessed the recruitment of the different GRK isoforms to the MOR (WT and STANT-2A mutant, as suggested). These data are now included in the manuscript (Figure 5, Table 4 and corresponding text in methods, results and discussion). Of note, while these data provide further insight into the mechanisms underlying MOR regulation, there are two potential caveats to their interpretation. First, overexpression of GRKs usually bypasses the endogenous modulation that typically results in barcode differences and ligand-dependent differences. We have previously shown that overexpression of GRK2, GRK3 and GRK5 alters the phosphorylation barcode of morphine such that it then resembles that of high efficacy agonists, including phosphorylation of Thr370 and Thr379 (Miess et al., 2018). Second, as GRK5 and GRK6 are membrane-tethered GRKs, measuring their ligand-induced recruitment to a GPCR, above the baseline signal, is challenging. Indeed, as shown in the new data provided, we detect a higher baseline BRET signal when expressing GRK5/6 vs GRK2/3, such that we do not detect a further increase upon agonist stimulation (Figure 5).

Having these caveats in mind, however, these experiments provide important information when comparing GRK2/3 recruitment to WT MOR vs STANT-2A MOR. As we have previously shown for GRK2, ligand-dependent GRK recruitment to MOR WT mirrors that of μ -arrestin 2 (Gillis et al., 2020). DAMGO and fentanyl display full agonism for GRK2 and GRK3 recruitment, while morphine displays less potent partial agonism (~50% maximal response compared to DAMGO). Interestingly, and similar to what we had observed with μ -arrestin 2 recruitment, GRK2 and GRK3 recruitment to the STANT-2A mutant does not reach the BRET levels detected for the WT receptor, suggesting that Thr376 and Thr379 are also required for the effective recruitment of GRK2 and GRK3. Altogether this suggests that these two residues have a dual role by i) facilitating GRK2/3 recruitment and, ii) upon phosphorylation promoting robust arrestin recruitment to MOR. These results add further texture to previously observed effects with more extensive receptor mutagenesis. Indeed, we have previously shown that mutation of all the Ser and Thr residues of the C-tail of MOR (11ST-A) reduced GRK2 recruitment, albeit inhibition was incomplete, and the remaining signal was naloxone sensitive (Miess et al., 2018). These data emphasise that Ser and Thr within the C-tail of MOR, while being substrates of GRKs, also participate in the initial engagement between GRKs and the receptor.

On that note, there is no structural information about the engagement mode of MOR with any GRK isoform. While structural information regarding the interactions of μ -arrestins (mostly μ -arrestin 1) with GPCRs is starting to emerge (Underwood et al., 2024), to-date, there is only two Cryo-EM structures of a GPCR in complex with GRKs, namely the rhodopsin-GRK1 and the NTSR1-GRK2 complexes (Chen et al., 2021; Duan et al., 2023). It is readily apparent that the arrestin and GRK complexes exhibit high conformational heterogeneity, which is likely a consequence of their characteristic ability to adapt and bind to hundreds of GPCRs. Indeed, even considering the caveats imposed by the experimental strategies used to achieve the structures of arrestin-GPCR complexes (which limit interpretation of phosphorylation barcodes), the available structural data highlight the versatility of arrestins and their ability to engage with GPCRs in dramatically different ways. The recent cryo-EM structures of Rho*-GRK1 and NTSR1-GRK2 demonstrate that the N-terminal end of the \pm N helix, highly conserved in all GRKs, directly inserts within the cytoplasmic cleft of the activated receptor. The Rho*-GRK1 structure suggests interactions of ICL1 loop and H8 of Rho* with GRK1. Interestingly, the basic residues in ICL1 and Arg8.51 in H8, are highly conserved in class A GPCRs, including the MOR. Based on the NTSR1-GRK2 complex structures, the extended loop of ICL3 or the elongated C-terminal tail of the GPCR can reach the active cleft of GRK2 and thus be available for phosphorylation by this kinase (in contrast to ICL1 and ICL2, unlikely to be accessible to the kinase active site). Given the data presented here, one can only speculate that the availability of more distal sites of the C-tail (including Thr376 and Thr379 of the MOR) to the active site of GRK5/6 is more limited, explaining their differential phosphorylation pattern when endogenously expressed. We have discussed these structural considerations in the revised manuscript.

- No statistical significance test has been performed on any data shown in the figures. If the authors want to establish that GRK2/3 or GRK5/6 play (or not) a role in the phosphorylation of a certain MOR residue, they need to provide accurate significance testing (indicating the test used and the number of independent experiments). For example, Fig. 1C needs to provide information if there is a significant difference between phosphorylation states in the control and KO cell lines.

This is particularly unclear for pS375 phosphorylation after morphine treatment, where the phosphorylation levels seem similar between GRK2/3 KO and GRK5/6 KO cells, despite the authors' conclusion that GRK5/6 (rather than GRK2/3) mediate pS375 phosphorylation.

We apologise for this oversight. We have performed statistical analyses in all our quantifications. For ease of interpretation, in Fig 1C we have stated the most relevant differences in the graph, namely the difference across KO cell lines vs control cell line and within a cell line vs control (vehicle) treatment. We have also included Supplementary Table 1 that provides the detailed analyses and modified the text accordingly.

- Data in Suppl. Fig. 1 shows ligand dependent G protein (mGsi) engagement with MOR in different GRK KO cell lines relative to control cells. The authors conclude the ability of MOR to interact with G proteins, as measured by recruitment of mGsi, was not affected in any of the knock-out cells. Yet no statistical testing is performed (e.g. for Emax) and contrary to the authors' comment, it appears that mGsi recruitment in control vs KO cells (e.g. S1A vs S1C) is strongly reduced.

We have now included Supplementary Table 2 which, like Table 1, shows the potency and maximal effect for each ligand in mGsi recruitment assays in the four cell lines. Statistical differences using unpaired t-tests of each cell line vs control cells are shown. The only statistical difference detected is the increase in Emax for fentanyl in GRK5/6 KO.

- The authors present data on rescued / enhanced β -arrestin2 recruitment upon reintroducing individual GRK family members into quadruple GRK2/3/5/6 KO cells. They draw conclusions on the ability of different GRKs to rescue / increase the β -arrestin2 engagement relative to control cells. To make this conclusion, the authors need to probe the level of GRK over-expression across all conditions. Are GRK2/3 more strongly expressed than GRK5/6? And how does the over-expression level compare to the endogenous GRK level of control cells?

In Drube et al (Drube et al., 2022) our team demonstrated that the relative expression of GRK isoforms remains unchanged upon individual or multiple knock-out of other isoforms. Importantly, we also showed that upon overexpression in iQ-GRK cells (using the same constructs and the same cells), the relative GRK expression compared to endogenous (control) levels varies between ~4-fold (GRK2), ~8-fold (GRK3 and GRK6) and ~15-fold (GRK5). Thus, it is unlikely that the effects observed upon GRK2 and GRK3 overexpression are due to higher expression compared to GRK5 and GRK6 (see Figure 1a, Supp Figure 1d and Supp Figure 3b and c in (Drube et al., 2022)).

Despite being overexpressed at potentially lower levels than the other GRK isoforms, we are convinced of GRK2 overexpression as we observe a well-established effect upon transfection in iQ-GRK cells, an increase in potency (leftward shift) and increased maximal effect for β -arrestin 2 recruitment of all ligands (see (Miess et al., 2018) for our previously published data or (Nickolls et al., 2013) for independent data).

- The authors need to accurately describe what the datapoints in the figures illustrate. For example, Suppl. Fig. 2A does not specify what the presented data points are, Suppl. Fig. 2B indicates ~3-10 independent experiments but shows exactly three datapoints for both conditions, Suppl. Fig. 3 indicates 3-10 independent experiments, yet the graph clearly plots more data points for each condition than 3-10.

This has now been clarified in the corresponding Figure Legend

- The authors need to provide more information on the plasmids / cDNA constructs used (in addition to citing the source papers). Are the authors working with receptor, arrestin, GRK constructs of human/mouse/canine/... origin? Also, how much DNA (receptor, arrestin, and GRK plasmids) is transfected in the β -arrestin2 recruitment assays?

The species of all relevant constructs and amounts of DNA transfected has been detailed in Materials and Methods.

Reviewer #2 (Remarks to the Author):

In this manuscript, Underwood et al. identified the roles of different GRK sub-families that can promote unique phosphorylation patterns on mu opioid receptor (MOR). The approach was

straightforward by using knockout cell lines and rescue strategies. The conclusion could be more convincing, and the work be more useful by addressing the following points.

Major comments:

1. The authors didn't explain the interesting observation that over-expression of single GRK could rescue and even enhance the Barr2 recruitment in the GRK knock-out cells. This could happen, possibly caused by different expression levels of GRKs, elevated levels of the phosphorylated MOR, changes in the phosphorylated barcode, or other potential factors?

The Reviewer raises a valuable reflection.

Different GRK expression levels: Unlikely, as commented in reply to Reviewer 1, our team has shown that upon overexpression in iQ-GRK cells (using the same constructs and the same cells), the relative GRK expression compared to endogenous (control) levels varies between ~4-fold (GRK2), ~8-fold (GRK3 and GRK6) and ~15-fold (GRK5) (see Figure 1a, Supp Figure 1d and Supp Figure 3b and c in (Drube et al., 2022)). Thus, it is unlikely that the differential effects observed upon GRK2 and GRK3 overexpression are due to higher expression compared to GRK5 and GRK6 (note that if overexpression underlies this effect, GRK6 would be expected to enhance recruitment beyond all the other GRKs, and this is not the case).

Elevated levels of phosphorylated MOR: Our previous work has shown that overexpression of GRK2, 3 and 5 in HEK293 increases the phosphorylation induced by morphine in residues readily phosphorylated in control cells (i.e. in the absence of GRK overexpression) (Doll et al., 2012). Thus, it is possible that for low efficacy agonists (that do not induce the levels of phosphorylation induced by higher efficacy ligands), overexpression of GRKs leads to increased phosphorylation. However, this is likely to be accompanied by changes in the phosphorylation barcode (see below).

Changes in the phosphorylation barcode: this is more evident, again, with low efficacy ligands that induce incomplete phosphorylation of the receptor. We have shown on several occasions that upon overexpression of GRK2, 3 and 5 in HEK293 cells, morphine's phosphorylation barcode is altered and resembles that of high efficacy agonists (Doll et al., 2011; Miess et al., 2018). Thus, it is likely that this change is also relevant for the increased β -arrestin 2 recruitment observed in Figure 3.

Other potential factors include an alteration of phosphorylation/dephosphorylation dynamics, which were beyond the scope of this work, but a future line of research.

However, it is also important to emphasise the main finding of this work, which is that Thr376 and Thr379 are key for the enhanced arrestin recruitment observed upon GRK overexpression, as when these residues are mutated, overexpression of neither GRK2 nor GRK5 does not enhance arrestin recruitment.

We thank the reviewer for prompting these reflections, which we have now incorporated in the Discussion section.

2. Based on the first comment, it is reasonable to speculate that the creation of distinct phosphorylation barcodes by individual endogenous GRKs may be attributed to variations in the expression levels of GRKs. Therefore, it is important to address the first comment and quantify the expression levels of endogenous GRKs.

As commented in reply to Reviewer 1, in Drube et al (Drube et al., 2022) our team demonstrated that the relative expression of GRK isoforms remains unchanged upon individual or multiple knock-out of other isoforms. In a parallel publication focused on the development of a quantitative assay to assess GRK expression levels, we showed that the HEK293 cells used here (HEK293 control) express similar amounts of GRK2 and GRK6 and approximately 5 times less GRK3 and GRK5 (Reichel et al., 2022). These data, together with the ligand-dependent effects, are incompatible with the creation of distinct phosphorylation barcodes solely due to the differential expression of GRK isoforms.

3. Although the authors identified GRK2/3 could phosphorylate all four sites and GRK5/6 only two sites, it is still unclear whether individual GRK promotes unique phosphorylation codes? For example, Do GRK2 and GRK3 target the same site? GRK5 and GRK6 on the same site as well? GRK5 and GRK6 appear to behave differently based on Figure 3. One experiment they could do is in the rescue experiments after introducing individual GRKs, they can determine the GRK-specific phosphorylation site.

The reviewer poses a relevant question; does each GRK isoform induce a unique phosphorylation

barcode? Unfortunately, testing this hypothesis still remains more challenging than anticipated. First, specific isoform selective KO in a control background shows very limited (if any) effects, likely reflecting the fact that other, even less efficacious kinases can access certain phosphosites, upon lack of competing effectors (Drube et al., 2022). Second, we and others have shown that overexpression of GRKs bypasses ligand-selective barcodes and results in a homogenous phosphorylation ((Miess et al., 2018) for MOR), limiting the interpretation of experiments run under such conditions. Even mutagenesis experiments are limited in interpretation as residues can contribute to recruitment as well as act as phosphosites themselves (see above). Thus, the path to evaluate the contribution of each isoform to ligand-dependent phosphorylation barcodes would be a triple KO strategy that results in the endogenous expression of a single GRK isoform. Only then, evaluation of phosphorylation barcodes would be informative of GRK-specific (and ligand-dependent) phosphorylation sites. Unfortunately, while the Hoffmann team is working on the generation of these cells, their use in the proposed experiments is beyond the current possibilities of our team.

4. It would be very useful to add beta-arrestin1 in the same functional test as beta-arrestin2 for comparison.

While the reviewer raises an interesting point, accumulated evidence shows that the extent of β -arrestin 1 recruitment to MOR is significantly lower than that of β -arrestin 2 (Drube et al., 2022; Miess et al., 2018). Apart from its transient interaction with β -arrestin 2, the different affinity for β -arrestin 1 vs β -arrestin 2 is a feature of MOR that classifies it as a Class A receptor in terms of GPCR-arrestin interaction (Groer et al., 2011; Janetzko et al., 2022; Oakley et al., 2000). This is particularly relevant in the case of morphine, where there is no detectable β -arrestin 1 recruitment upon endogenous GRK expression. Thus, as the current work focused on agonist-dependent effects upon GRK KO, we did not embark in the assessment of β -arrestin 1 recruitment.

5. In line 192, why do morphine in GRK2/3 knockout cells recruit beta-arrestin2 5-fold higher than the wt cells?

Indeed, as the Reviewer mentions, in contrast to the effects observed for DAMGO and fentanyl, the relative E_{max} for morphine in β -arrestin 2 recruitment in GRK2/3 cells is increased while the phosphorylation data show that in these cells MOR phosphorylation is restricted to Ser375, and to a lesser extent to Thr370 (Figure 1). We hypothesise that this effect could be explained by different mechanisms. First, as mentioned in the discussion, it must be considered that by eliminating the expression of a certain protein, the competition between mediators of the same effect is severely altered; namely, that if the main effector is knocked-out, the action of other, less efficient effectors may become more apparent. I.e., one can suggest that in the absence of GRK2/3, other kinases (e.g GRK5/6) have less competition for the same sites in terms of effectiveness of the kinase activity or, alternatively, in terms of spatial rearrangement around the receptor (steric hindrance). In addition, other kinases may also benefit from the absence of GRK2/3 and phosphorylate sites outside the 370TREHPSTANT379 motif. In this context, PKC is an obvious candidate, as it has been shown to modify MOR in a morphine-dependent manner, and different than GRK2 (Bailey et al., 2006; Kelly et al., 2008). We have recently generated isoform-family selective PKC CRISPR-KO cells expressing MOR and will test this hypothesis in the future. Finally, the removal of a GRK2/3-mediated inhibition on MOR/ β -arrestin 2 recruitment cannot be discarded, although the identification of such inhibition is beyond the current work. We have expanded these reflections in the Discussion section.

Minor points:

6. While authors have already done comprehensive statistical analysis throughout the manuscript, there are still some areas lacking this analysis, such as Figure 1C. For Table 2, the authors claimed that statistical analysis has been performed, but there are no corresponding labels provided.

We apologise for these oversights. We have performed statistical analyses in all our quantifications. For ease of interpretation, in Fig 1C we have stated the most relevant differences in the graph namely the difference across cell lines vs control cell line and within a cell line vs control (vehicle) treatment. We have also included a Supplementary Table (Supp Table 1) that provides the detailed analyses and modified the text accordingly. Table 2 now includes the statistical differences within the table.

7. Many GPCRs have phosphorylation sites in the intracellular loop regions. There are a few Ser/Thr residues located in the ICLs of MOR. The authors didn't look at them or is there no contribution from the ICL regions? This could be discussed.

The reviewer is correct, in addition to the 11 residues in the C-tail of MOR, there are 5 potential phosphorylation sites in the ICLs of the receptor: Thr97 in ICL1, Thr180 in ICL2/cytoplasmic part of TM4, and Ser261, Ser266 and Ser268 in ICL3/cytoplasmic parts of TM5 and TM6. Additionally, these sites are conserved between mouse and human receptors. We have put some effort into assessing these sites, however, there are two important limitations that hamper our progress in this area. First, we still lack selective antibodies that recognise such sites, and second, Ala substitution mutagenesis experiments have rendered mutant receptors with impaired cell surface expression, limiting the interpretation of subsequent concentration-response data.

Reviewer #3 (Remarks to the Author):

The manuscript "Key phosphorylation sites for robust β -arrestin2 binding at the MOR revisited" uses an innovative set of experiments with CRISPR-KO cell lines, phosphosite-antibodies and site-directed mutagenesis to clearly show phosphorylation patterns of μ opioid receptor (MOR) agonists involved in the recruitment of β -arrestin. Importantly the studies compare phosphorylation patterns with endogenous levels of GRK 2,3,5,6 with complete KO of all GRKs, as well as with over-expression of each single GRK on the complete KO background. The manuscript is very clearly written with appropriate attention to previous studies and their caveats. The data presented are based on appropriate numbers of replicates with appropriate controls. Overall, the manuscript is an important contribution to the field and indicates that tissue expression and localization of GRK complements will play a critical role in determining the desensitization, internalization and trafficking of MORs. Although other studies have proposed this, the studies in this manuscript for the first time clearly link GRK-mediated phosphorylation patterns to ligand-dependent differences in MOR recruitment of β -arrestin2.

The only confusing statement in the manuscript is the sentence (lines 193-195) that recruitment of mGsi was not affected in any of the knock-out cells (Supp Fig 1). The figure shows differences in recruitment but there are no statistical analyses, thus it is not clear what the basis of the statement is.

We have now included Supplementary Table 2 which, like Table 1, shows the potency and maximal effect for each ligand in mGsi recruitment assays in the four different cell lines. Statistical differences using unpaired t-tests of each cell line vs control cells analysis are shown. The only statistical difference detected is the increase in Emax for fentanyl in GRK5/6 KO.

Reviewer #4 (Remarks to the Author):

In this study, Underwood and collaborators aimed to identify key phosphorylation sites in the μ opioid receptor (MOR) involved in β -arrestin2 recruitment upon some opioid agonist. All the study was realized in a heterologous system (HEK293 cells transfected with different constructs). Using HEK293 cell lines knocked down for GRKs, phosphosite specific antibodies and site-directed mutagenesis, they demonstrated that the GRK subfamilies (GRK2/3 or GRK5/6) differentially contribute to ligand-induced phosphorylation barcodes and β -arrestin2 recruitment. The paper is very well written and the experiments were well conducted and suitable to answer the scientific question.

However, I have some remarks and questions:

- In the abstract: I suggest removing the term of "desensitisation" as no data on desensitisation are provided in the study.

We respectfully disagree with the reviewer here, the term "desensitisation" is only used in the first sentence of the abstract to provide context to our work, the abstract does not state any intention to show desensitisation data.

- In the material and methods :

"In the HA-MOR construct, is it mouse MOR ?

Yes, indeed it is the mouse MOR, this is now stated more clearly

â€¢ The authors used different MOR constructs (HA-MOR, Flag-mMOR-NLuc). But, did they check that the ligands used in the study bind and activate these different MOR constructs in the same manner? It is important to link phosphorylation studies and beta-arrestin2 recruitment studies.

The author raises a valid and crucial concern. Through our multiple publications together and separately, we have shown that the addition of N- and C-terminal tags to the MOR does not alter its pharmacology. For example, in Gillis et al (Gillis et al., 2020), we used a collection of assays that utilised N- and C- terminally tagged MOR, and that recapitulated the pharmacology of all the tested ligands.

â€¢ Regarding the statistical analysis, the authors mentioned â€œStatistical significance was determined using an unpaired T-Test, corrected for multiple comparisons using a Holm-Å ÅdÅjk test (*: p Å 0.05).â€ I am a bit confused here. Indeed, T-test are used to compared two groups and the Holm-Å ÅdÅjk correction for multiple test is used after an ANOVA, used to compare more than 2 groups. So, this must be clarified by mentioning which statistical analysis was used in each experiment and the subsequent results. For instance, in the case of ANOVA, F statistic should be given along with results of the multiple comparisons. Moreover, did the authors verify the normality of the distribution as well as the homoscedasticity of the variance ? If yes, which test did they use ? Finally, it is mentioned that â€œData show the mean Å± SEM of at least 5 independent experiments performed in triplicateâ€ but in some figures, it is mentioned â€œ3 independent experimentsâ€.

We apologise for this confusion; all the points above have been carefully considered and addressed in the corresponding sections; Materials and Methods and Figure and Table legends.

In addition, we can confirm that the data used for statistical comparisons (i.e pEC50s and Emax and band intensity in phosphorylation blots) are normally distributed (see (Christopoulos, 1998) for reference).

â€¢ For the phosphorylation experiments, how the authors choose the agonist concentrations?

To assess receptor phosphorylation, we selected saturating concentrations of agonists. These saturating concentrations were determined through our extensive research on opioid receptor pharmacology (Just et al., 2013).

- In figure 1:

â€¢ Fig 1B: the np-MOR bands look more intense in control cells. Did the authors quantify receptor expression in these different cell lines?

In Drube et al., (Drube et al., 2022) we show that there is no difference in receptor levels between control and KO cell lines. Previous studies show that the levels of expression in these cells are ~2300 fmol/mg memb protein.

â€¢ Fig 1C: I am bit surprised that no SEM appeared on DAMGO groups. Even if it was normalized to DAMGO, SEM should appeared. Why no statistical analysis was performed?

We apologise for these oversights. We have performed statistical analyses in all our quantifications. For ease of interpretation, in Fig 1C we have stated the most relevant differences in the graph namely the difference across cell lines vs control cell line and within a cell line vs control (vehicle) treatment. We have also included a Supplementary Table (Supp Table 1) that provides the detailed analyses and modified the text accordingly. Table 2 now includes the statistical differences within the table.

For this graph, data was DAMGO in control cells was used for normalisation (100%), to facilitate comparison within cell line as well as across cell lines (our focus). Thus, DAMGO in control cells is the only data set with no associated error. All the other DAMGO groups are presented as mean Å± sem.

- In figure 3 and table 2 : The recruitment of beta-arrestin2 appears to vary depending on whether GRK2 or GRK3 is overexpressed. How could it be explained?

Reviewer 2 rightly commented on the lack of statistical symbols in Table 2. This has now been amended. There are no differences between the potencies of Emax upon GRK2 or GRK3 overexpression.

References

- Bailey, C. P., Smith, F. L., Kelly, E., Dewey, W. L., & Henderson, G. (2006). How important is protein kinase C in mu-opioid receptor desensitization and morphine tolerance? *Trends Pharmacol Sci*, 27(11), 558-565. <https://doi.org/10.1016/j.tips.2006.09.006>
- Chen, Q., Plasencia, M., Li, Z., Mukherjee, S., Patra, D., Chen, C. L., Klose, T., Yao, X. Q., Kossiakoff, A. A., Chang, L., Andrews, P. C., & Tesmer, J. J. G. (2021). Structures of rhodopsin in complex with G-protein-coupled receptor kinase 1. *Nature*, 595(7868), 600-605. <https://doi.org/10.1038/s41586-021-03721-x>
- Christopoulos, A. (1998). Assessing the distribution of parameters in models of ligand-receptor interaction: to log or not to log. *Trends Pharmacol Sci*, 19(9), 351-357. [https://doi.org/10.1016/s0165-6147\(98\)01240-1](https://doi.org/10.1016/s0165-6147(98)01240-1)
- Doll, C., Konietzko, J., Poll, F., Koch, T., Holtt, V., & Schulz, S. (2011). Agonist-selective patterns of micro-opioid receptor phosphorylation revealed by phosphosite-specific antibodies. *Br J Pharmacol*, 164(2), 298-307. <https://doi.org/10.1111/j.1476-5381.2011.01382.x>
- Doll, C., Páll, F., Peuker, K., Loktev, A., Gláck, L., & Schulz, S. (2012). Deciphering μ -opioid receptor phosphorylation and dephosphorylation in HEK293 cells. *British journal of pharmacology*, 167(6), 1259-1270. <https://doi.org/10.1111/j.1476-5381.2012.02080.x>
- Drube, J., Haider, R. S., Matthees, E. S. F., Reichel, M., Zeiner, J., Fritzwanker, S., Ziegler, C., Barz, S., Klement, L., Filor, J., Weitzel, V., Kliewer, A., Miess-Tanneberg, E., Kostenis, E., Schulz, S., & Hoffmann, C. (2022). GPCR kinase knockout cells reveal the impact of individual GRKs on arrestin binding and GPCR regulation. *Nat Commun*, 13(1), 540. <https://doi.org/10.1038/s41467-022-28152-8>
- Duan, J., Liu, H., Zhao, F., Yuan, Q., Ji, Y., Cai, X., He, X., Li, X., Li, J., Wu, K., Gao, T., Zhu, S., Lin, S., Wang, M. W., Cheng, X., Yin, W., Jiang, Y., Yang, D., & Xu, H. E. (2023). GPCR activation and GRK2 assembly by a biased intracellular agonist. *Nature*, 620(7974), 676-681. <https://doi.org/10.1038/s41586-023-06395-9>
- Gillis, A., Gondin, A. B., Kliewer, A., Sanchez, J., Lim, H. D., Alamein, C., Manandhar, P., Santiago, M., Fritzwanker, S., Schmiedel, F., Katte, T. A., Reekie, T., Grimsey, N. L., Kassiou, M., Kellam, B., Krasel, C., Halls, M. L., Connor, M., Lane, J. R., . . . Canals, M. (2020). Low intrinsic efficacy for G protein activation can explain the improved side effect profiles of new opioid agonists. *Sci Signal*, 13(625). <https://doi.org/10.1126/scisignal.aaz3140>
- Groer, C. E., Schmid, C. L., Jaeger, A. M., & Bohn, L. M. (2011). Agonist-directed Interactions with Specific β -Arrestins Determine μ -Opioid Receptor Trafficking, Ubiquitination, and Dephosphorylation. *The Journal of biological chemistry*, 286(36), 31731-31741. <https://doi.org/10.1124/pr.111.004598>
- Janetzko, J., Kise, R., Barsi-Rhyne, B., Siepe, D. H., Heydenreich, F. M., Kawakami, K., Masureel, M., Maeda, S., Garcia, K. C., von Zastrow, M., Inoue, A., & Kobilka, B. K. (2022). Membrane phosphoinositides regulate GPCR-beta-arrestin complex assembly and dynamics. *Cell*. <https://doi.org/10.1016/j.cell.2022.10.018>
- Just, S., Illing, S., Trester-Zedlitz, M., Lau, E. K., Kotowski, S. J., Miess, E., Mann, A., Doll, C., Trinidad, J. C., Burlingame, A. L., von Zastrow, M., & Schulz, S. (2013). Differentiation of Opioid Drug Effects by Hierarchical Multi-Site Phosphorylation. *Molecular pharmacology*, 83(3), 633-639. <https://doi.org/10.1124/mol.112.082875>
- Kelly, E., Bailey, C. P., & Henderson, G. (2008). Agonist-selective mechanisms of GPCR desensitization. *Br J Pharmacol*, 153 Suppl 1(S1), S379-388. <https://doi.org/10.1038/sj.bjp.0707604>
- Miess, E., Gondin, A. B., Yousuf, A., Steinborn, R., Mässllein, N., Yang, Y., Gäldner, M., Ruland, J. G., Bänemann, M., Krasel, C., Christie, M. J., Halls, M. L., Schulz, S., & Canals, M. (2018). Multisite phosphorylation is required for sustained interaction with GRKs and arrestins during rapid μ -opioid receptor desensitization. *Science signaling*, 11(539). <https://doi.org/10.1126/scisignal.aas9609>
- Nickolls, S. A., Humphreys, S., Clark, M., & McMurray, G. (2013). Co-Expression of GRK2 Reveals a Novel Conformational State of the μ -Opioid Receptor. *PLoS one*, 8(12), e83691. <https://doi.org/10.1371/journal.pone.0083691.t003>
- Oakley, R. H., Laporte, S. A., Holt, J. A., Caron, M. G., & Barak, L. S. (2000). Differential affinities of visual arrestin, beta arrestin1, and beta arrestin2 for G protein-coupled receptors delineate two major classes of receptors. *J Biol Chem*, 275(22), 17201-17210. <https://doi.org/10.1074/jbc.M910348199>
- Reichel, M., Weitzel, V., Klement, L., Hoffmann, C., & Drube, J. (2022). Suitability of GRK Antibodies for Individual Detection and Quantification of GRK Isoforms in Western Blots. *Int J Mol Sci*, 23(3). <https://doi.org/10.3390/ijms23031195>
- Underwood, O., Haider, R. S., Sanchez, J., & Canals, M. (2024). Arrestin-centred interactions at the membrane and their conformational determinants. *Br J Pharmacol*. <https://doi.org/10.1111/bph.16331>

Reviewer comments:

Reviewer #1

(Remarks to the Author)

The authors have improved the manuscript. The new data on GRK recruitment adds novelty to the work. Specific comments are listed below:

- Figure 5: the legend for 5A appears inaccurate. The data presented not only show baseline BRET measurements but also BRET upon agonist addition. This should be mentioned in the legend and the authors must indicate the agonist concentration shown here. Also, according to the data presented in 5A, recruitment of GRK2 and GRK3 to WT or STANT-2A receptors appear very similar.
- I am confused by the following statement (first results paragraph): "When the MOR was expressed in cells lacking GRK2/3 (deltaGRK2/3), there was no difference between the phosphorylation pattern induced by morphine, DAMGO or fentanyl. Thus, upon endogenous expression of GRK5/6 isoforms, all ligands induced phosphorylation of Thr370 and Ser375 to a similar extent, while no phosphorylation of Thr376 or Thr379 could be detected (Fig. 1)." The quantified data in Fig. 1C shows that in deltaGRK2/3 cells, morphine no longer promotes significant Ser375 phosphorylation (non-significant when compared to vehicle control). Thus, how can the authors conclude that GRK5/6 mediate phosphorylation of MOR at Ser375 following morphine?
- It is difficult to properly see Fig. 1C, including the significance testing. It would be advised to increase the panel size / adapt to a larger font in Fig. 1C.
- The new table 2 summarizes data shown in Suppl. Fig. 1, and for analyses the authors have set Emax of DAMGO to 100% for each cell line (wt and GRK KO cell lines). Yet, if the authors want to hold on to their following statement "As expected, the ability of MOR to interact with G proteins, as measured by recruitment of mGsi, was not significantly affected in any of the knock-out cells (Supp Fig 1)" they need to test how much GRK KO affects the Emax (deltaBRET) of mGsi recruitment across the different cell lines. As mentioned in my previous comments, mGsi recruitment to MOR in control cells (S1A) appears much higher (20%) than in deltaQ GRK (S1B) and deltaGRK2/3 cells (S1C). The authors need to adapt their analyses to test this.

Reviewer #2

(Remarks to the Author)

The authors have well addressed my questions. I support the publication of the manuscript.

Reviewer #4

(Remarks to the Author)

The authors responded to many of my questions. However, some points still need to be clarified:

- The authors cited a reference (Christopoulos, 1998) regarding the normal distribution of data but did they actually check their data which test did they use ?
- Concerning the level of MOR in the different cell lines, the authors cited Drube et al. (2022) study to claim there was no difference in the receptor level in the different cell lines. Did they refer to the supplementary figure 2a of these cited paper ? If yes, it seems that some GRK KO cell lines have a reduced level of MOR, moreover I do not think I saw any MOR quantification among the different cell lines. This point should be at least consider in the discussion.
- I did not get the answer regarding the normalisation of the data in fig. 1C. I'm very sorry but I still don't understand why there was no SEM in DAMGO group. Here's an example of normalisation, which I think is probably more accurate (but I'm maybe wrong). Values in the control group: 142, 85, 150, 90, so the mean is 116,75. After normalisation to the control itself, values are : 121.627409, 72.80513919 , 128.4796574, 77.08779443, so the mean if 100 and SD if 29.116. I think this point is important, because the normalisation methods will determine the results of the statistical analysis. In the Drube et al. (2022) mentioned by the authors, the results of MOR phosphorylation in supplementary fig 2 seemed to be normalized to control and yet SEM were calculated.
- Unless I missed these information, I did not find the F statistic in the different ANOVA tests. It should be mentioned as well as the results of interaction in the case of a two-way ANOVA.

Author Rebuttal letter:

Reviewers' comments (COMMSBIO-24-0176)

Reviewer #1 (Remarks to the Author):

The authors have improved the manuscript. The new data on GRK recruitment adds novelty to the work. Specific comments are listed below:

- Figure 5: the legend for 5A appears inaccurate. The data presented not only show baseline BRET measurements but also BRET upon agonist addition. This should be mentioned in the legend and the authors must indicate the agonist concentration shown here. Also, according to the data presented in 5A, recruitment of GRK2 and GRK3 to WT or STANT-2A receptors appear very similar.

Indeed, the data in these panels are not only baseline measurements, they include the signal measured upon incubation with DAMGO, morphine and fentanyl at 10 μ M. Figure legend has been edited. The message of these panels is that due to increased baseline BRET signals for GRK5 and GRK6, the response of agonists cannot be detected, hence the inclusion of agonist responses.

In terms of the GRK2 and GRK3 recruitment to WT and STANT-2A, while the raw data may look indeed similar, the agonist-induced response, upon baseline subtraction is not. This is illustrated in the concentration response curve in Figure 5C. The figure below shows the 10 μ M response when baseline-corrected:

- I am confused by the following statement (first results paragraph): "When the MOR was expressed in cells lacking GRK2/3 (deltaGRK2/3), there was no difference between the phosphorylation pattern induced by morphine, DAMGO or fentanyl. Thus, upon endogenous expression of GRK5/6 isoforms, all ligands induced phosphorylation of Thr370 and Ser375 to a similar extent, while no phosphorylation of Thr376 or Thr379 could be detected (Fig. 1)." The quantified data in Fig. 1C shows that in deltaGRK2/3 cells, morphine no longer promotes significant Ser375 phosphorylation (non-significant when compared to vehicle control). Thus, how can the authors conclude that GRK5/6 mediate phosphorylation of MOR at Ser375 following morphine?

The phosphorylation induced by morphine in deltaGRK2/3 cells is not different from vehicle in that cell line, as stated by the reviewer. However, that phosphorylation is also not different to the phosphorylation induced by morphine in the control cell line. While this is likely due to the variability in that set of data, we have edited the confusing statement to make it more accurate.

- It is difficult to properly see Fig. 1C, including the significance testing. It would be advised to increase the panel size / adapt to a larger font in Fig. 1C.

Panel 1C has been enlarged, and the font for significance made larger. We are happy to work with the editorial team if this is not clear upon publication.

- The new table 2 summarizes data shown in Suppl. Fig. 1, and for analyses the authors have set Emax of DAMGO to 100% for each cell line (wt and GRK KO cell lines). Yet, if the authors want to hold on to their following statement "As expected, the ability of MOR to interact with G proteins, as measured by recruitment of mGsi, was not significantly affected in any of the knock-out cells (Supp Fig 1)" they need to test how much GRK KO affects the Emax (deltaBRET) of mGsi recruitment across the different cell lines. As mentioned in my previous comments, mGsi recruitment to MOR in control cells (S1A) appears much higher (20%) than in deltaQ GRK (S1B) and deltaGRK2/3 cells (S1C). The authors need to adapt their analyses to test this.

We assume the reviewer is referring to Suppl Table 2. Note that in that table we have already included the data that the reviewer asks for. The second row, for each ligand, has the Emax expressed as deltaBRET as well as a % of the control cell line in brackets. We have performed statistical tests on these data, and the only significant increase on mGsi recruitment was detected for fentanyl in deltaGRK5/6 cells. Thus, our statement is correct.

Reviewer #2 (Remarks to the Author):

The authors have well addressed my questions. I support the publication of the manuscript.

Thank you

Reviewer #4 (Remarks to the Author):

The authors responded to many of my questions. However, some points still need to be clarified:
"The authors cited a reference (Christopoulos, 1998) regarding the normal distribution of data but did they actually check their data which test did they use?"

Due to the N value of our experiments, normality of data was assessed using the Shapiro-Wilk test in GraphPad Prism v10. pEC50s, Emax and band intensity were normally distributed. This is now mentioned in the Methods section "Data analysis".

"Concerning the level of MOR in the different cell lines, the authors cited Drube et al. (2022) study to claim there was no difference in the receptor level in the different cell lines. Did they refer to the supplementary figure 2a of these cited paper? If yes, it seems that some GRK KO cell lines have a reduced level of MOR, moreover I do not think I saw any MOR quantification among the different cell lines. This point should be at least consider in the discussion."

The reviewer is correct, we are referring to Supp Fig2 in Drube et al. Please see below the quantification of the total MOR levels presented in that figure:

I did not get the answer regarding the normalisation of the data in fig. 1C. I'm very sorry but I still don't understand why there was no SEM in DAMGO group. Here's an example of normalisation, which I think is probably more accurate (but I'm maybe wrong). Values in the control group: 142, 85, 150, 90, so the mean is 116,75. After normalisation to the control itself, values are : 121.627409, 72.80513919 , 128.4796574, 77.08779443, so the mean is 100 and SD is 29.116. I think this point is important, because the normalisation methods will determine the results of the statistical analysis. In the Drube et al. (2022) mentioned by the authors, the results of MOR phosphorylation in supplementary fig 2 seemed to be normalized to control and yet SEM were calculated.

This has now been reanalysed to address this reviewer's comments, SEM for DAMGO in control cell line is shown.

Unless I missed these information, I did not find the F statistic in the different ANOVA tests. It should be mentioned as well as the results of interaction in the case of a two-way ANOVA.

This information is now provided in Supp Table 1 and where applicable.

Version 2:

Reviewer comments:

Reviewer #1

(Remarks to the Author)

Thank you for clarifying these points. My comments have been addressed by the authors.

The statement: 'We show that both GRK2/3 and GRK5/6 subfamilies promote phosphorylation of 370Thr and 375Ser, with morphine-induced phosphorylation of these residues specifically mediated by GRK5/6, while DAMGO and fentanyl can engage all GRKs to promote such phosphorylation' (see abstract) is not supported by the data shown. The data in Fig. 1 (and statistically analysed in Fig. 1C) clearly show that GRK2/3 KO leads to a strong decrease in morphine-driven 370Thr and 375Ser phosphorylation. In GRK2/3 KO cells, no significant phosphorylation is detected in response to morphine (comparison vehicle vs morphine). So GRK2/3 must play a role in the morphine-driven response similar to what is the case for DAMGO and fentanyl.

Reviewer #4

(Remarks to the Author)

I thank the authors for the improvements they have made to their manuscript. I still have three minor points:

- Supplementary Figure 5: The control group is presented with SEM. It should be calculated (see the remark I made regarding normalization).
- Supplementary Table 1: The F statistic is still missing (for instance: $F(x,y) = ??, p < 0.05$).
- The authors mentioned they used two-way ANOVA, but I did not see it applied (though I may have missed it). They could consider removing this mention from the Materials and Methods section.

Author Rebuttal letter:

Response to Referees (COMMSBIO-24-0176)

Reviewer #1 (Remarks to the Author):

Thank you for clarifying these points. My comments have been addressed by the authors.

The statement: 'We show that both GRK2/3 and GRK5/6 subfamilies promote phosphorylation of 370Thr and 375Ser, with morphine-induced phosphorylation of these residues specifically mediated by GRK5/6, while DAMGO and fentanyl can engage all GRKs to promote such phosphorylation' (see abstract) is not supported by the data shown. The data in Fig. 1 (and statistically analysed in Fig. 1C) clearly show that GRK2/3 KO leads to a strong decrease in morphine-driven 370Thr and 375Ser phosphorylation. In GRK2/3 KO cells, no significant phosphorylation is detected in response to morphine (comparison vehicle vs morphine). So GRK2/3 must play a role in the morphine-driven response similar to what is the case for DAMGO and fentanyl.

This reviewer is correct in the vagueness of these statements. Indeed, we cannot discard a GRK2/3-mediated phosphorylation of 370Thr or 375Ser upon morphine stimulation. All sections (including the Graphical Abstract) have been revised to consider this.

Reviewer #4 (Remarks to the Author):

I thank the authors for the improvements they have made to their manuscript. I still have three minor points:

â€¢ Supplementary Figure 5: The control group is presented with SEM. It should be calculated (see the remark I made regarding normalization).

SEM for the normalised control has been added

â€¢ Supplementary Table 1: The F statistic is still missing (for instance: $F(x,y) = ??$, $p < 0.05$).

I sincerely apologise for this oversight; I uploaded the wrong file which did not have the F statistics. I have corrected this now.

â€¢ The authors mentioned they used two-way ANOVA, but I did not see it applied (though I may have missed it). They could consider removing this mention from the Materials and Methods section.

The reviewer is correct, the mention has been removed from Materials and Methods.
